# Single-molecule RNA sizing enables quantitative analysis of alternative transcription termination

Gerardo Patiño-Guillén[1], Jovan Pešović[2], Marko Panić[2,3], Dušanka Savić-Pavićević[2], Filip Bošković [1]✉ & Ulrich Felix Keyser [1]✉

Transcription, a critical process in molecular biology, has found many applications in RNA synthesis, including mRNA vaccines and RNA therapeutics. However, current RNA characterization technologies suffer from amplification and enzymatic biases that lead to loss of native information. Here, we introduce a strategy to quantitatively study both transcription and RNA polymerase behaviour by sizing RNA with RNA nanotechnology and nanopores. To begin, we utilize T7 RNA polymerase to transcribe linear DNA lacking termination sequences. Surprisingly, we discover alternative transcription termination in the origin of replication sequence. Next, we employ circular DNA without transcription terminators to perform rolling circle transcription. This allows us to gain valuable insights into the processivity and transcription behaviour of RNA polymerase at the single-molecule level. Our work demonstrates how RNA nanotechnology and nanopores may be used in tandem for the direct and quantitative analysis of RNA transcripts. This methodology provides a promising pathway for accurate RNA structural mapping by enabling the study of full-length RNA transcripts at the single-molecule level.

The process of converting information from DNA into RNA is known as transcription and it is of fundamental need in gene expression[1]. Transcription is performed by the enzyme RNA polymerase (RNAP), which binds to a recognition sequence in DNA known as a promoter. RNAP unwinds double-stranded DNA and uses the template DNA strand and base pairing complementarity with the DNA template to create an RNA copy. Eventually, RNAP reaches a termination sequence, where it dissociates from the DNA template and the RNA transcript is released[2].

Transcription is not limited to the conversion of linear DNA templates. RNAPs may also transcribe circular DNA in a rolling circle manner[3–6]. Rolling circle transcription involves the continuous unidirectional synthesis of RNA from circular DNA, producing consecutive copies of the DNA sequence. The number of consecutive copies in RNA depends on how many times the polymerase goes around the DNA[7].

Besides their biological relevance, RNAPs have demonstrated immense technological impact due to their high fidelity and processivity[8], which enables the production of high quality and high yield in vitro transcribed RNA, used in RNA therapeutics and RNA vaccines[9,10], for example.

For the development of RNA technologies and proficient characterization of gene expression, an in-depth understanding of transcript diversity is required. An accurate assessment of transcript size and heterogeneity is needed, as well as precise identification of splicing patterns and transcript variants. The main approaches to studying RNA are based on bulk techniques such as gel electrophoresis, quantitative reverse transcription–polymerase chain reaction (qRT-PCR), and RNA sequencing (RNA-seq)[11]. Nevertheless, these techniques may face limitations in understanding RNA diversity in its native form. qRT-PCR requires additional enzymatic steps and may encounter reverse transcription and polymerase biases[12]. The same

[1]Cavendish Laboratory, University of Cambridge, Cambridge, UK. [2]University of Belgrade – Faculty of Biology, Centre for Human Molecular Genetics, Belgrade, Serbia. [3]Institute of Virology, Vaccines and Sera "Torlak", Belgrade, Serbia. ✉e-mail: fnb24@cam.ac.uk; ufk20@cam.ac.uk

happens with RNA-seq, which may also face short-read limitations[13]. Despite its practical simplicity, accurate analysis of agarose gels can be complicated as no information about the sequence is obtained and electrophoretic mobility can be affected by the conformation of RNA[14]. Unexpected transcription species in gels may indicate transcription initiation in an alternative promoter sequence or they may correspond to the same RNA product with another conformation and, therefore, different electrophoretic mobility. Considering the capabilities of current RNA analysis methods, a robust characterization platform is needed, which can provide an accurate description of transcript heterogeneity and retains the native diversity of RNA transcripts.

Single-molecule approaches have been employed to study both RNA transcripts and RNAPs themselves. Previously, the spatial distribution of RNAP has been characterized in growing bacteria using single-molecule fluorescence microscopy[15,16]. By combining optical tweezers and fluorescent microscopy, the dynamics of RNAP at each step of transcription has been investigated[17], showing that single-molecule techniques can provide further insight into the characterization of transcription[18].

Nanopore sensing is another versatile single-molecule technology. It relies on the application of a potential through a nanoscale orifice in an electrolytic solution to produce an ionic current[19–21]. As analytes pass through the pore, driven by electrophoretic force, ions are depleted, causing ionic current blockage associated with the analyte's volume, conformation, or size[22]. The technique enables single-molecule detection and has facilitated the study of a wide range of biological species, including DNA[23,24], proteins[19,25,26] antibodies[19,27], RNAs[28–31], ribosomes[32], and viruses[33,34]. Recently, DNA-based labeling of RNA has been performed, allowing for RNA structural mapping with nanopores[35]. This strategy involves hybridizing RNA with complementary DNA oligonucleotides to produce an RNA–DNA duplex. The short DNA strands may have overhangs to attach sequence-specific labels that can be distinguished by nanopore current signals.

In this work, we propose a method to study single-molecule transcription of T7 RNA polymerase (T7RNAP) in vitro with nanopores and RNA nanotechnology. Our strategy enables the quantitative description of the T7RNAP's ability to continuously transcribe different DNA templates and identifies premature transcription termination sites using sequence-specific labeling. The strategy prevents reverse transcription biases, it circumvents the need for pre-amplification and allows to characterize transcripts with a length of 10 s of kilobases. We first transcribe a linear DNA that contains a T7RNAP promotor sequence and that lacks a terminator sequence. After transcription with T7RNAP, each RNA product is hybridized with complementary DNA oligos to produce an RNA–DNA duplex named RNA identifier (RNA ID) which enables parallel single-transcript mapping with nanopore sensors. Our strategy identifies premature transcription termination of T7RNAP in the linear DNA construct within the origin of replication (OriC) sequence. We next employ a circular DNA construct with the same backbone sequence as the linear DNA template to assess the rolling circle transcription of T7RNAP. Since no termination sequence was introduced to our construct, we anticipated transcription termination at random positions and at random time points. However, after the identification of a premature transcription termination site in the OriC, we explore the enzyme's processivity in this DNA construct and study the correlation between transcript length and the premature termination sequence.

By coupling nanopore sensing with RNA nanotechnology, we demonstrate structural mapping of diverse populations of RNA transcripts, hence revealing premature terminator sequence in OriC. Our approach unlocks future study of the processivity of enzymes and the resulting sequence landscape of transcribed RNA.

## Results

### RNA identifiers enable quantitative analysis of transcription termination in linear DNA

We start with the analysis of the transcripts from a linear DNA construct which contains a promoter sequence for T7RNAP (Fig. 1a). The promoter mediates transcription initiation by T7RNAP. The linear DNA construct was produced from a circular DNA plasmid by restriction digestion using DraIII. The restriction site is upstream of the T7RNAP promoter (Fig. 1a; more details of the restriction digestion sequence position are shown in Supplementary Table 1). The construct is 3.1 kilobase pair-long DNA and it was engineered to have the OriC, 12 CTG tandem repeats downstream of the T7RNAP promoter, and importantly, no termination sequence. The CTG repeats are located downstream of the T7RNAP promoter, and the OriC is in the middle of the construct (Fig. 1a; the exact distances are illustrated in Supplementary Table 1). Due to their vicinity to the promoter, CTG tandem repeats are used as a reference to indicate the position of transcription initiation and to indicate the 3′ to 5′ directionality of transcripts.

Our linear DNA construct was in vitro transcribed with T7RNAP. Since no termination sequence was introduced into the construct, the enzyme should transcribe the entire linear DNA and detach from it once it reaches its end (Fig. 1a). T7RNAP was engineered to be a processive enzyme; hence we expect low dissociation without a defined terminator sequence[8]. After performing in vitro transcription, transcribed RNA was analysed using agarose gel electrophoresis (Fig. 1b). DNA and RNA ladders (lanes 1 and 2, respectively) allowed us to assess the running speeds of DNA and RNA, respectively. In lanes 3 and 4, we run transcribed RNA without and with DNase I treatment, respectively.

Surprisingly, RNA that was not treated with DNase I (lane 3) produces three bands in the agarose gel. The topmost band is ascribed to linear DNA used as a template, which is ~3 kbp long. The chemical nature of this band is confirmed by DNase I treatment (lane 4), which removes the band. DNase I treatment does not remove the middle and lower bands which suggests they correspond to RNA products. The middle band, which is slightly lower than 3 kbp RNA, could be ascribed to RNA where T7RNAP transcribes the entire DNA construct until it reaches its end, which should be 2.9 kb long (Fig. 1b). The bottom band, however, located between 1 and 2 kb in the RNA ladder, is unexpected. This RNA could originate from transcription initiation in an alternative promoter sequence, it could correspond to RNA of an alternative transcript variant, or it could be the same RNA product with another conformation and, therefore, different electrophoretic mobility. Unraveling the identity of this RNA species from the gel is challenging or impossible, since gel electrophoresis does not provide any information related to the sequence of the RNA molecule.

Our technique allows the identification of transcript variants resulting from alternative transcript processing solely based on the DNA sequence of the gene. In order to elucidate the identity of these RNA species we use RNA nanotechnology and nanopore sensing. Initially, we hybridize RNA with short complementary single-stranded DNA (ssDNA) oligonucleotides (oligos) to produce RNA IDs (Fig. 1c)[35]. RNA IDs recognize structural elements of RNA sequence, allowing us to determine where transcription begin, ends, and any other structural arrangements. Through the inclusion of labeled oligos, RNA IDs facilitated the characterization of RNA transcripts by nanopore sensing, which confirmed two types of RNA after transcription. In the first transcript type, the polymerase reaches the end of the linear DNA template (END RNA), as expected. In the second RNA type, nanopore readout shows that premature transcription termination occurs in the middle of the DNA (PT RNA, Fig. 1c), thus originating the unexpected RNA species.

The RNA IDs were formed with oligos complementary to the linear DNA sequence (oligos sequences in Supplementary Table 2), to perform simultaneous sequence-specific labeling of all RNA transcripts in the sample, independently of the position where the polymerase

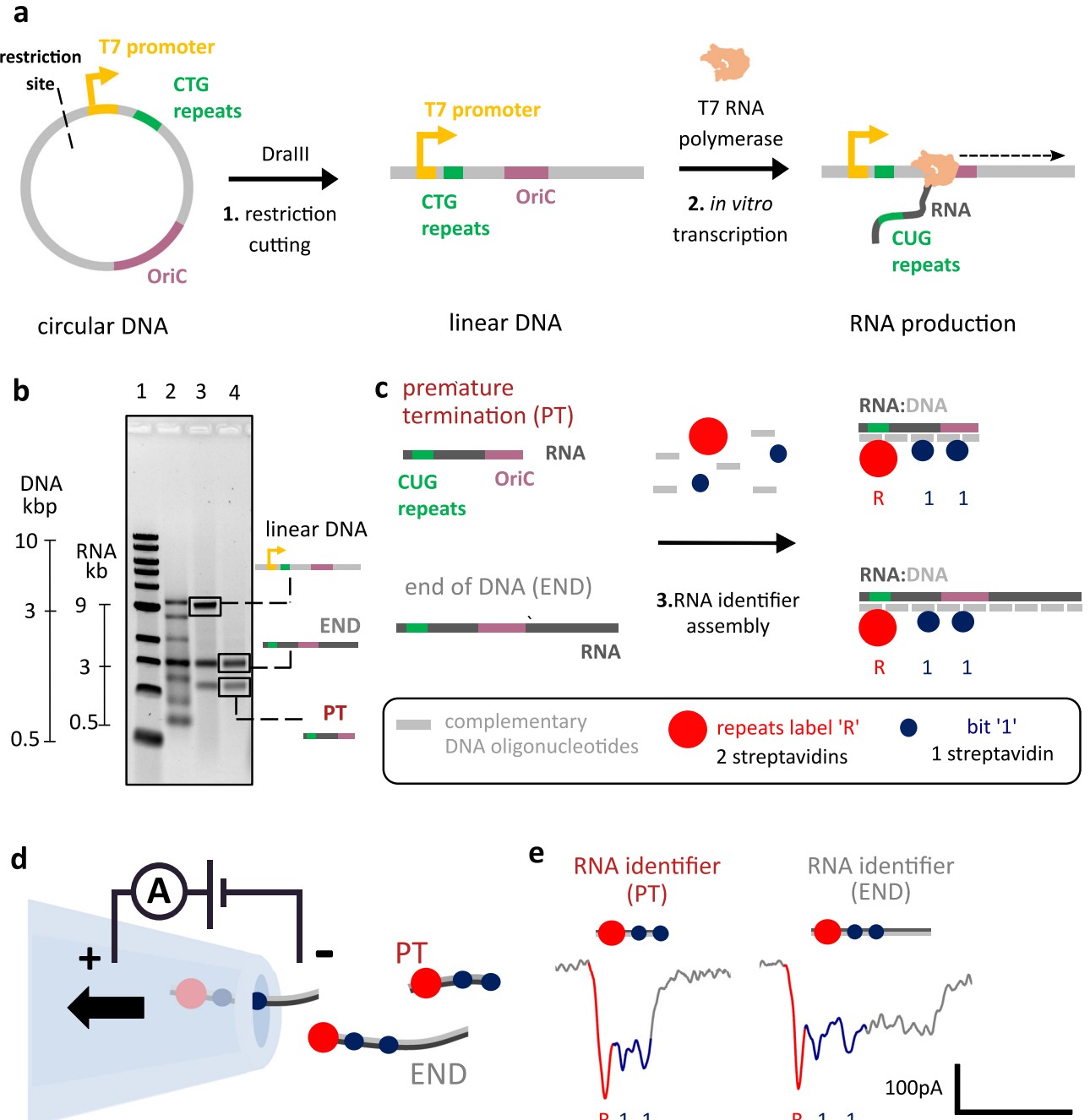

**Fig. 1 | RNA identifier (ID) assembly enables single-molecule structural analysis of RNA transcripts using nanopore sensing. a** In vitro transcription of linear DNA containing 12 CTG tandem repeats. A circular DNA construct contains a single T7RNAP promoter, the OriC, 12 CTG tandem repeats, and a DraIII restriction site. The circular DNA construct was linearized using DraIII, by cutting upstream of the T7RNAP promoter. The linear DNA was in vitro transcribed using T7RNAP. **b** Agarose gel electrophoresis of transcribed linear DNA indicates the presence of two RNA species (lane 4), one in which the RNA polymerase transcribes the entire linear DNA (END) and RNA of unknown nature. With nanopore sensing, it is later revealed that the lower band corresponds to RNA, where transcription is prematurely terminated within the OriC sequence (PT). Gel lanes: 1 – 1 kbp ladder; 2 – single-stranded RNA ladder; 3 – transcribed RNA from linear DNA; 4 - transcribed RNA from linear DNA treated with DNase I. Transcriptions were performed in triplicate. **c** RNA transcripts were hybridized with short complementary DNA

oligonucleotides (~40 nt), producing RNA IDs. CUG repeats in RNA were labeled with two streptavidins (repeats label "R"), indicating the beginning of transcription given their vicinity to the promoter. Further positioning along the transcript is achieved using bits "1" that have one streptavidin enabling their distinction from label "R". **d** RNA IDs made from RNA transcripts that have different termination points (PT or END) translocating through the nanopore. Nanopore readout is based on electrophoretically driven transport of negatively charged RNA IDs through the nanopore towards a positively charged electrode. **e** Nanopore RNA ID readouts for PT and END RNAs are presented left and right, respectively. RNA ID for both PT and END shows downward spikes associated to the labeled repeats "R" (red) and "1" bits (blue). PT translocation finishes right after the second "1" bit hence making a clear distinction towards END translocations, which takes longer to translocate through the pore. Source data are provided as a Source Data file.

began or stopped transcribing. Two types of labels have been designed to enable the mapping of individual transcripts. Given their proximity to the promoter, the CUG tandem repeats in RNA (originally CTG in DNA) were used as a label to identify the position where transcription was initiated. RNA IDs can pass through the nanopore in both 5′– 3′ and 3′– 5′ directions, the repeats can be used as a marker to identify the direction in which the molecule translocated through the pore. These repeat labels are named "R" (Fig. 1c and Supplementary Fig. 1) and they are constituted of a $(CAG)_{10}$ oligo with an overhang sequence containing a biotin group at its 3′ end (docking strand). This oligo is reverse-complementary to the CUG repeats in target RNA. A complementary strand to the docking strand (imaging strand) also has a 3′ biotin. The docking strand and imaging strand can bind two monovalent streptavidins[36] creating a single ionic current drop in nanopore signals.

Besides, parts of the RNA are decorated with "1" bits (Fig. 1c). These lack the biotin group in the docking strand, which only allows the binding of one streptavidin per bit and produces a distinguishable signal from the CUG repeats "R" (Supplementary Fig. 1). More details about the design and assembly of "R" labels and "1" bits are shown in Supplementary Fig. 2.

RNA ID assembly facilitates single-molecule transcript analysis using glass nanopore sensors (Fig. 1d). After applying a voltage bias, negatively charged RNA IDs translocate through the nanopore towards the positively charged electrode, blocking the pore partially and inducing a transient drop in ionic current. The current signal is directly associated with the RNA ID design and produces a unique molecular fingerprint of the RNA ID that passes through the nanopore[35].

The translocation of RNA IDs produces current signatures with downward spikes from the labeled repeats "R" and "1" bits included in the design. The current spikes are generated by the streptavidin-biotin conjugate, which blocks more ions than the RNA ID backbone on its own, producing a deeper ionic current drop[35]. "R" labels contain two biotin-streptavidin conjugates, thus blocking twice the current in comparison to "1" bits (Fig. 1e; "R" – red' and "1" bits – blue) that contain one streptavidin. The position of the labels in the RNA ID produces a characteristic ionic current readout, which provides an additional level of specificity to exclude false-positive detection. Hence, we can exclude that contaminants create such events as demonstrated previously[34,35].

Two RNA IDs were detected, originating from two different RNAs. In the first one, T7RNAP reaches the end of DNA (END RNA ID). In the second, transcription termination is caused by a premature termination sequence (PT RNA ID). The RNA IDs for both PT and END transcripts are electrophoretically driven through the same nanopore, enabling characterization of the transcript products while preventing nanopore readout variability[37]. In END RNA IDs, the nanopore translocation shows the "R" and "1" bit current spikes, followed by a prolonged plateau which is attributed to the region with no streptavidin labels in the RNA ID, only constituted by RNA–DNA duplex. The plateau correlates to the region spanning from the OriC sequence to the end of the linear DNA template. This readout corresponds to RNA where transcription terminated after the enzyme most likely detached at the end of the DNA template. The second type of current signature detected, PT RNA ID, also presents "R" and "1" bit spikes, meaning it shares the same initial sequence as the full-length transcript, however, the translocation ends right after the second '1' bit current spike and no plateau is observed. This "1" bit is located 0.2 kbp into the OriC, indicating that this RNA species originates from premature transcription termination within the OriC sequence.

Having performed structural mapping of RNA to identify a premature transcription termination position, we next sought to conduct a quantitative analysis of the transcripts. The RNA IDs of both termination points (PT and END, shown in Figs. 1e, 2a) are quantitatively studied using nanopores by looking at two parameters for each RNA ID translocation. First, we compute the translocation time, i.e., the time it

takes a single RNA ID molecule to pass through the pore, and second, the charge deficit of the translocation event, which represents the surface area of the nanopore event[38–40] (Fig. 2b). Raw PT and END RNA ID events are shown in Supplementary Fig. 3 and Supplementary Fig. 4, respectively. Translocation time and charge deficit of all translocation events detected are shown in Supplementary Figs. 5, 6.

Translocation time and charge deficit of 70 unfolded (linear) events of each transcript type (140 events in total) are displayed in a scatter plot (Fig. 2c). The histogram of translocation time and charge deficit are also plotted above the $x$ and $y$ axis, respectively. The plot shows that RNA IDs from END transcripts have longer translocation times and larger charge deficits than RNA IDs from PT transcripts. Longer RNA IDs, in this case, RNA IDs from END transcripts, require more time to translocate through the nanopore. As they take longer to translocate, more ions are depleted, which makes the charge deficit larger. Translocation time and charge deficit have a linear dependency (Fig. 2c). Additionally, the histograms of charge deficit and translocation time exhibit distinct distributions for PT and END RNA IDs, demonstrating that both parameters are suitable for the identification of transcripts with different termination sites.

We next performed single-molecule sizing of the transcripts by calculating an estimated length of each transcript in base pairs from their respective nanopore readout (Fig. 2d). In the RNA ID design, the labeled repeats "R" are located 0.5 kbp away from its adjacent "1" bit, and two "1" bits are separated 0.6 kbp from each other. A base pair-to-time conversion factor for each individual event is obtained by associating the base pair distance between each label to the distance between translocation times of their current spikes in the nanopore events. The overall event translocation time is then translated into a base pair estimate using the conversion factor for each RNA ID event.

RNA ID translocation times were converted to base pairs, revealing two distinct distributions, summarized in Fig. 2e. The PT-type RNA IDs had a mean length of $(1.75 \pm 0.17)$ kbp. This finding is in agreement with the position of the OriC within the linear DNA construct (Supplementary Table 1). The RNA IDs from END transcripts had a mean length of $(3.19 \pm 0.27)$ kbp, which is also in agreement with the length of the linear DNA available for transcription after digestion with DraIII. Variability in transcript sizing due to changes in translocation speed along an RNA ID molecule[41] are within error. Transcription termination detected in nanopore sensors agrees with gel electrophoresis assays which show ~47% termination efficiency (Supplementary Figs. 7, 8).

## RNA identifiers characterize T7 RNA polymerase processivity in rolling circle transcription

Thanks to the molecular profiling capabilities of our strategy, we perform single-molecule analysis of T7RNAP transcription in a more complex scenario, using the circular DNA construct introduced in the previous section. The construct is not linearized, and preserves one T7RNAP promoter, the OriC sequence, and CTG tandem repeats downstream of the T7RNAP promoter. Supplementary Table 3 shows the oligos used for RNA ID assembly of transcripts produced from this construct.

Since supercoiled DNA can present a roadblock for in vitro transcription (Supplementary Fig. 9), we performed relaxation to the circular DNA to facilitate transcription[42,43]. The relaxation was done with *Escherichia coli* Topoisomerase I and subsequently in vitro transcribed with T7RNAP (Fig. 3a). DNase I treatment is shown in Supplementary Fig. 10. As the DNA construct is circular and lacks a termination sequence, the enzyme should not necessarily fall off after the transcription of the whole DNA sequence. Instead, the polymerase can transcribe the circular construct several times continuously, in a rolling circle manner and transcription termination is governed solely by T7RNAP stochastic dissociation or by potential alternative termination sequences, such as previously mentioned OriC (Fig. 1). After the polymerase transcribes the entire construct once, it can continue to

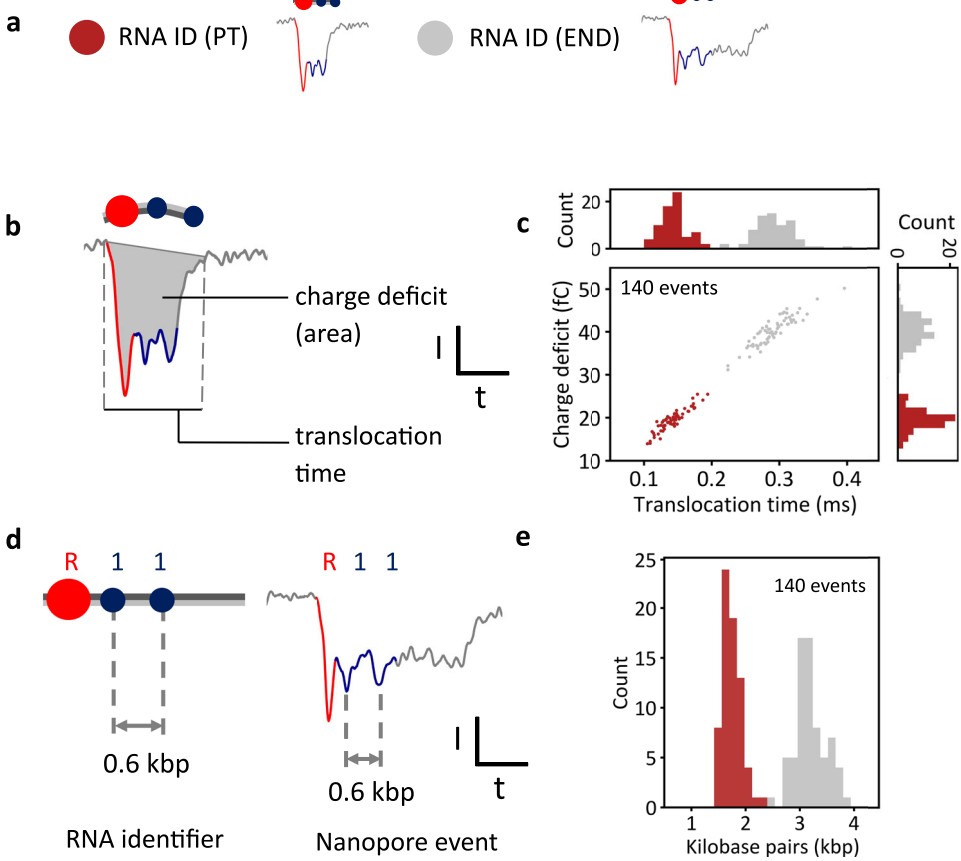

**Fig. 2 | Quantitative analysis of single-molecule sizing of RNA determines the position of the alternative terminator. a** Example nanopore events showing translocations of RNA IDs from PT RNAs (red), where transcription terminated prematurely, and RNA IDs from long END RNAs (gray), where transcription terminated at the end of the linear DNA. **b** Physical parameters that are used to characterize nanopore translocation events, including event charge deficit that represents the area of an event, and translocation time. **c** Scatter plot of charge deficit against translocation time for RNA IDs of PT (red) and END (gray) RNA transcripts, which shows the linear dependence of both parameters. END RNA IDs require more time to translocate through nanopores. These block the ionic current for a longer time, depleting more ions, and producing a larger charge deficit.

Histograms of translocation time and charge deficit show distinct distributions between both transcripts. Datapoints correspond to the translocation of RNA IDs measured within the same nanopore. The sample size was 140. **d** The ID design is used to convert translocation time into an estimate of the RNA length in base pairs, by knowing the base pair distance between labels and associating that distance to the time difference between the current downward spikes of the labels. **e** Base pair length of all molecules converted from translocation time (in *c*), which shows two distinct distributions. PT distribution has a mean length of $(1.75 \pm 0.16)$ kbp and END transcripts have a mean length of $(3.19 \pm 0.27)$ kbp. Errors correspond to standard deviation. Source data are provided as a Source Data file.

transcribe the construct's sequence on numerous occasions without stopping[3], producing an RNA molecule with multiple copies of the construct's sequence (Fig. 3b).

In our DNA construct, the CTG repeats (CUG repeats in RNA) are located only 74 nucleotides away from the T7RNAP promoter. By labeling the CUG repeats in RNA, we use the "R" repeats label to identify every time T7RNAP goes past the promoter. If T7RNAP starts transcribing the circular DNA for the first time, the 'R' repeats label will be detected once, however, if T7RNAP transcribes the whole sequence of the circular DNA and transcribes beyond the promoter, a second "R" repeats label will appear. Corresponding numbers of labels will be detected if T7RNAP transcribes past the promoter on a third or fourth occasion. In this work, we refer to each instance we see the "R" repeats label, as a transcription cycle '*n*'. A transcription cycle "*n*" corresponds to every time T7RNAP starts transcribing the circular DNA construct from the promoter. For the first transcription cycle ($n = 1$), only one repeats label "R" will be detected, and it will indicate the position where transcription is initiated. The subsequent appearance of the repeats label "R" indicates the beginning of a new transcription cycle ($n = 2$). Additionally, "1" bits were incorporated as done for RNA IDs of the linear construct, to facilitate nanopore readout and perform single-molecule mapping.

Example nanopore events of RNA IDs with $n = 1$ and $n = 2$ are shown in Fig. 3b. RNA IDs from one transcription cycle ($n = 1$), in which the whole circular DNA sequence was transcribed once, produce a nanopore current signature with the downward spikes from labeled repeats "R" and two current spikes from two "1" bits. When two full transcription cycles ($n = 2$) are completed by T7RNAP, the nanopore readout represents double the current signature of $n = 1$. Hence two sets of R' peaks and bits can be observed (Fig. 3b) in the event with twice the duration. The sequence-specific labels of RNA IDs facilitate the distinction of the position within the RNA ID in which the second transcription cycle began and the position where transcription terminated.

With our strategy, we observed RNA IDs originating from one (blue), two (orange), three (green), four (red), or five (purple) transcription cycles (Fig. 3c), showcasing capability of T7RNAP to synthesize transcripts in the kilobase range[44]. Each nanopore readout shows its corresponding "R" and "1" current downward spikes, which can be used to infer the number of transcription cycles. Nanopore-identified RNA transcripts with multiple transcription cycles were also visualized by agarose gel electrophoresis (Fig. 3d, lane 1), using an ssRNA ladder as a reference to infer transcript length (lane 2). The multiple bands in lane 1 are ascribed to RNA products from one (blue) to five (purple)

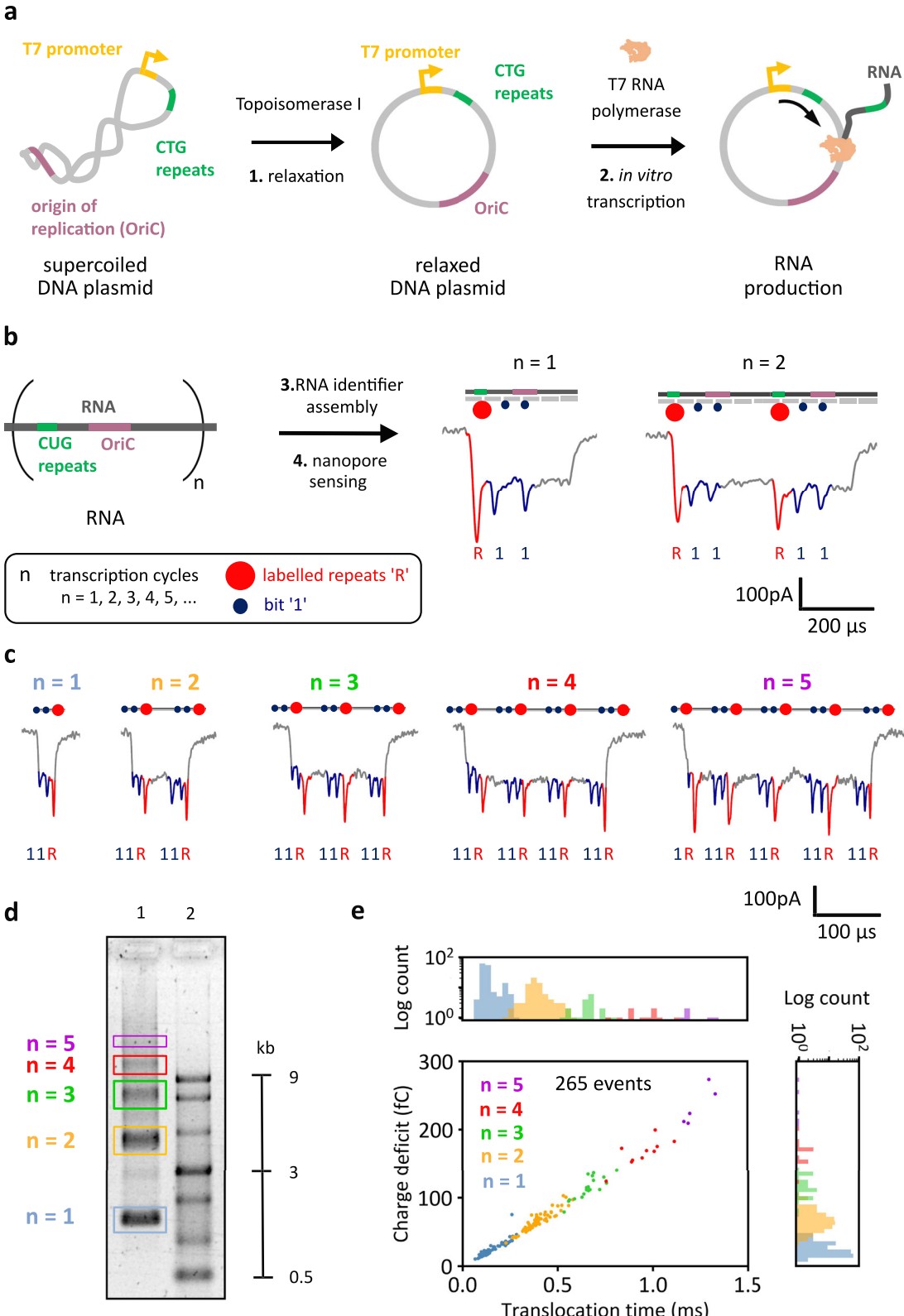

transcription cycles. RNA IDs from produced transcripts were also characterized using agarose electrophoresis and presented the multiple bands associated with the different transcription cycles (Supplementary Fig. 11).

For an exemplary nanopore measurement, the obtained translocation events were categorized by the number of transcription cycles (the sample size was 265 unfolded RNA ID events). As can be observed from Fig. 3e, translocation events of larger RNA IDs, produced from more transcription cycles, have larger translocation times, because longer molecules take more time to translocate through the pore. For the same reason, the charge deficit increases in events associated with larger transcript sizes (Fig. 3e). Both the translocation time and charge deficit are seen to be linearly correlated. Single-molecule quantification of RNA with different transcription cycles reveals that the

**Fig. 3 | RNA identifier (ID) assembly enables single-molecule characterization of transcription of circular DNA. a** In vitro transcription of the DNA plasmid. The supercoiled plasmid contains the T7RNAP promoter, the OriC, and 12 CTG tandem repeats. The DNA was relaxed with Topoisomerase I and in vitro transcribed using T7RNAP in a rolling-circle manner. **b** T7RNAP produces RNA that contains multiple ("$n$") copies of the plasmid sequence. Transcribed RNA was hybridized with complementary DNA oligonucleotides (~40 nt), producing RNA IDs. CUG repeats in RNA were labeled (repeats label "R"), indicating the beginning of a transcription cycle given their vicinity to the T7RNAP promoter and "1" bits were included to facilitate transcript identification. Nanopore RNA ID readouts ($n = 1$ and $n = 2$) show downward spikes associated with the labeled repeats "R" (red) and "1" bits (blue). **c** Nanopore readouts for RNA IDs of transcripts produced from 1 to 5 transcription cycles "$n$", each showing the labeled repeats "R" and "1" bits. More example events are shown in Supplementary Figs. 12–16. RNA IDs can translocate both 5'–3' and 3'–5' directions through the nanopore. These events show translocations in the 3'–5' direction. Events translocating in the opposite direction are shown in Supplementary Fig. 16. **d** Electrophoretic mobility shift assay shows a single-stranded RNA ladder (ssRNA) on lane 2. Lane 1 shows transcription products of the relaxed plasmid, confirming nanopore readout of various transcript lengths. Gel: 1% (w/v) agarose, 1 × TBE, 0.02% sodium hypochlorite. Transcription was performed in triplicate. **e** Scatter plot of charge deficit against translocation time for RNA IDs of multiple transcription cycles, which shows the linear dependence of both parameters. Events with more cycles correspond to longer RNA molecules that need more time to translocate through nanopores, which blocks the ionic current for longer. Datapoints correspond to the translocation of RNA IDs measured within the same nanopore. Histograms plotted in linear scale are presented in Supplementary Fig. 17. The sample size was 265. Exemplary measurement comparing charge deficit of unfolded (linear) and folded RNA IDs is presented in Supplementary Fig. 18. Source data are provided as a Source Data file.

transcript count decreases with the number of transcription cycles, meaning their length. This can also be observed in histograms of charge deficit and translocation time (Fig. 3e). Both histograms show localized time distributions, each corresponding to a transcription cycle, which indicates that there are RNA IDs with a similar length for each of the transcription cycles.

Up to this point, two experimental procedures have been discussed. First, the study of premature transcription termination of T7RNAP in a linear DNA template through single-molecule sizing of the transcripts using the RNA ID design. Second, the assembly of RNA IDs from transcripts with multiple lengths, produced from the rolling circle transcription of a circular DNA, and their identification using nanopore sensing.

Both procedures are now combined to elucidate T7RNAP's premature transcription termination in the circular plasmid and to obtain further understanding of the enzyme's capability to continuously synthesize RNA without releasing the template strand. During transcription of the plasmid, premature termination can occur at the OriC, as previously discussed, or the polymerase can continue transcribing past this sequence (Fig. 4a). If the polymerase continues transcribing, it can perform another transcription cycle and can go around the plasmid until it encounters the premature termination sequence again or falls off after multiple transcription cycles. It is then expected that most of the RNA products are the result of termination at the OriC despite the number of transcription cycles that the polymerase performs. Representative nanopore translocation events of RNA IDs with termination within the OriC sequence are shown in Fig. 4b, where the translocation finishes after the second '1' bit regardless of the number of transcription cycles.

The translocation time of the nanopore events was normalized into base pairs using the ID design (Fig. 4c), which showed localized distributions attributed to premature termination within the OriC for multiple transcription cycles (Fig. 4d). $n = 1$ events show RNA IDs of ~1.6 kbp, suggesting premature termination within the OriC right after initiation of the first transcription cycle. $n = 2$ transcript size is ~4.6 kbp, which corresponds to the transcription of the full plasmid plus the transcription from the T7RNAP promoter to the OriC in the second transcription cycle. The same occurs with $n = 3$ events, these have a transcript size of ~7.9 kbp, which corresponds to two full transcription cycles and the transcription from the T7RNAP promoter to the OriC in the third cycle. Premature transcription termination at the multiple transcription cycles was further confirmed via agarose gel electrophoresis in DNA constructs engineered with the OriC at different positions from the T7RNAP promoter. Supplementary Figs. 22, 23 show transcription of linear and circular DNA with insertions of different lengths between the T7RNAP promoter and OriC, and Supplementary Fig. 24 shows transcription of a different linear DNA template that also contains a T7RNAP promoter and OriC. The sequences of these constructs are presented in Supplementary Tables 4, 5.

Nanopore measurements were performed to further validate the detection of transcription termination in OriC (Supplementary Fig. 25). The oligos used to assembly RNA IDs for these transcripts are shown in Supplementary Table 6.

The less prominent distributions, located in between the main populations associated with premature termination, at ~3.1 kbp for $n = 1$ and ~5.9 kbp for $n = 2$, are ascribed to dissociation of the T7RNAP. These agree with the faint bands in agarose gels of the same transcript sizes presented in Fig. 3d.

The normalized length distribution of the RNA IDs after performing molecular sizing shows a decay-like behavior (Supplementary Fig. 26), meaning that there are fewer RNA molecules resulting from a higher number of transcription cycles. Our results indicate the expected behavior, as the RNA polymerase must overcome the premature termination sequence in each of the cycles. Hence, the RNA ID length distribution describes T7RNAP's capability to continuously transcribe circular DNA and the efficiency of alternative termination.

## Discussion

We showcase the utilization of RNA IDs to analyse complex transcript populations at the single-molecule level. This approach offers a means to directly label RNA, enabling the concurrent analysis of multiple transcripts and the visualization of sequence-specific biomarkers. RNA IDs can be engineered to target genes of interest. Hence, we can produce labels to identify which regions of the gene's sequence is retained at the transcript level. By doing so, we address the shortcomings of commonly employed transcript characterization methods like gel electrophoresis, which can be challenging to interpret and lack sequence information. Furthermore, our method overcomes reverse transcription and DNA polymerase biases and is not restricted to the analysis of short transcripts.

Transcription of linear DNA and subsequent RNA ID assembly are utilized to investigate premature transcription termination. The translocation time and the charge deficit are reliable parameters for describing the transcript length. Moreover, an estimated base pair length for different transcripts was computed without additional reference molecules. This enabled the identification of a premature transcription termination site within the OriC (~47% termination efficiency) in our linear DNA construct.

We conclude premature transcription termination from the location of the second "1" bit in the RNA ID design, which is positioned 0.2 kbp downstream of the beginning of the OriC sequence. The "1" bit produces a current spike at the end of the translocation event, indicating that rho-independent termination occurs downstream of the "1" bit. A sequence (Supplementary Table 7) downstream and in proximity to the "1" bit shows structural similarity to T7RNAP terminators previously reported[45], which makes it a potential candidate responsible for the reported termination. Previously, termination efficiency of up to 92% has been shown for bacterial RNA polymerases[46,47], indicating

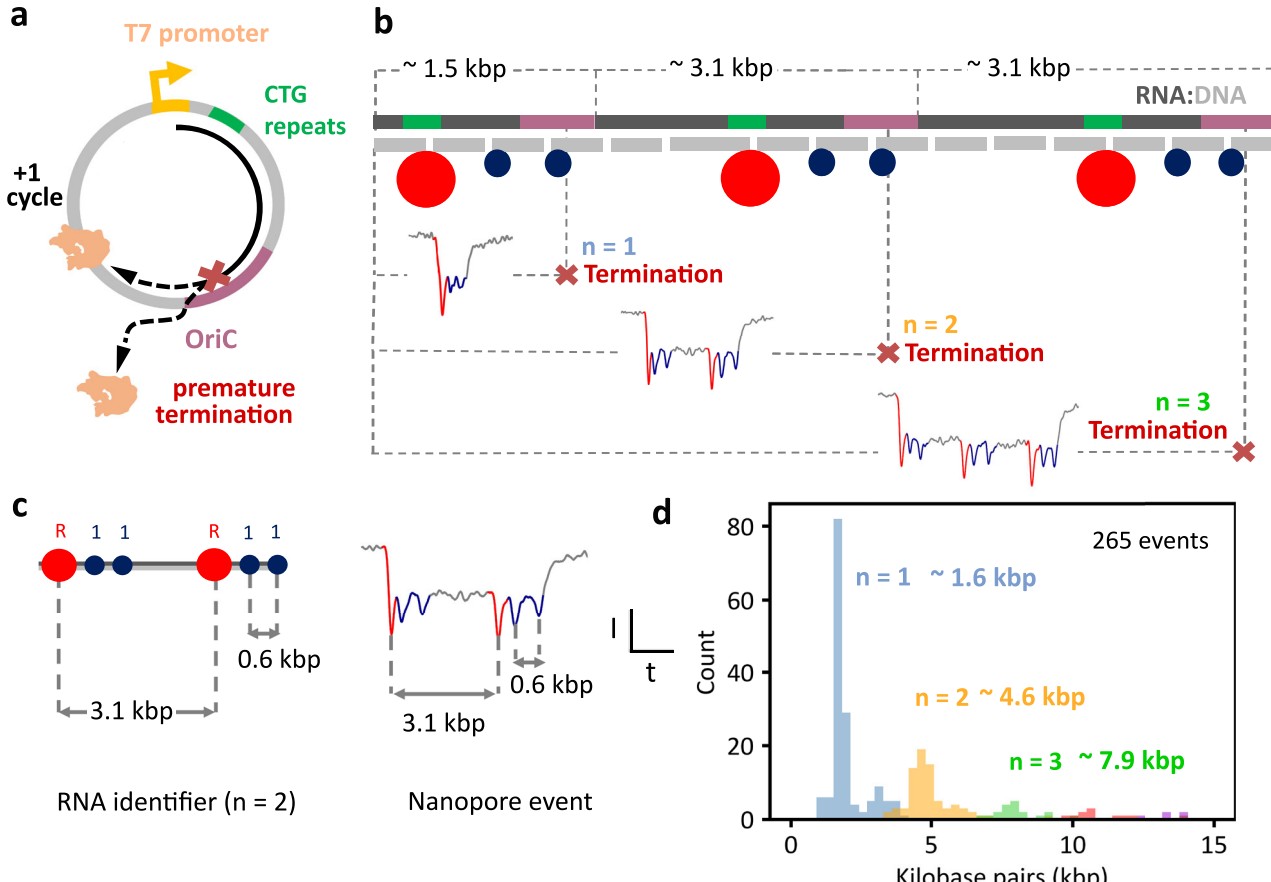

**Fig. 4 | T7 RNA Polymerase's transcription capability and premature termination analysis in circular DNA at the single-molecule level. a** Schematic of DNA plasmid, showing how termination can occur at the OriC sequence or the polymerase can continue transcribing the circular DNA for another cycle. **b** Schematic of RNA ID showing example events of termination after 1, 2, or 3 transcription cycles. **c** The ID design is used to convert translocation time into an estimate of the RNA length in base pairs. The distance between current spikes of "R" and "1" bits is translated to base pairs to obtain a conversion factor to obtain the length in base pairs for the whole event. **d** Histogram of normalized length (base pairs) of RNA IDs produced from the transcription of circular DNA, showing premature termination at -1.6, -4.6, and -7.9 kbp in the OriC sequence, after transcription of 1 to 3 cycles, respectively. The sample size was 265 nanopore events. Raw translocation events are presented in Supplementary Figs. 19–21. Source data are provided as a Source Data file.

termination in the OriC is not as efficient for T7RNAP. The OriC also affected T7RNAP's capability to continuously transcribe the circular DNA construct, where we observed that termination within the OriC dominates at different transcription cycles with a similar termination efficiency as observed for the linear construct.

Transcription termination of bacterial RNA polymerases in OriC has been identified in multiple bacterial systems[46]. For this reason, exploring the effect of the origins of replication in transcription could offer valuable insights into the understanding of transcript diversity.

The refined understanding we provide here is essential for downstream applications, including the synthesis and characterization of therapeutic RNAs and messenger RNA vaccines, as well as for in vitro and in vivo production of RNAs and proteins. Moreover, the comprehensive tools here presented are useful for the study of RNA produced from rolling circle transcription, which finds therapeutic applications in the enhancement of RNA delivery in cells and the prolongation of protein expression[5,48,49]. RNA ID assembly coupled with nanopore sensing characterization can also be used for examining transcription kinetics.

Finally, the strategy presented is also well-suited for describing transcription processes with relevance in fundamental research, such as analysis of gene expression and profiling, or for the identification of transcription regulation factors for DNA-based enzymes. Our work also indicates that single-molecule characterization of biomarkers in RNA, like labeled CUG tandem repeats are suitable for the identification of transcription initiation. Successful transcription and recognition of these repeats, which have mutation-induced capabilities[50,51], opens a door towards in-depth characterization of tandem repeats, which may have clinical relevance[50,52].

## Methods

### Circular DNA plasmid production

The circular plasmid was chosen from the existing plasmid collection from the Centre for Human Molecular Genetics, University of Belgrade-Faculty of Biology, which was previously produced by several reactions of cloning and subcloning of CTG repeats originating from human *DMPK* locus using pJet1.2/blunt cloning vector from CloneJET PCR Cloning Kit (Thermo Fisher Scientific). The subcloned region sequence was verified with Sanger sequencing before prior use. The whole plasmid sequencing was performed by the DNA sequencing facility, Department of Biochemistry, University of Cambridge. The plasmid was propagated in *Escherichia coli* JM110 strain upon chemical transformation. The plasmid DNA was purified using GeneJET Plasmid Miniprep Kit (Thermo Fisher Scientific) and eluted in nuclease-free water. The quality of plasmid DNA was checked using agarose gel electrophoresis, while the concentration was measured using a Qubit 2.0 fluorometer (Thermo Fisher Scientific).

### Circular DNA plasmid propagation

JM110 strain of *Escherichia coli* was used for propagation of circular plasmid with CTG repeats. About 100 ng of plasmid was mixed with

50 μL of competent cells (0,085 M CaCl$_2$, 15% glycerol). The mixture was incubated on ice for 30 min prior to heat shock at 42 °C for 90 s. The cells were turned back on ice for 5 min and spread on LB agar plates supplemented with ampicillin (0.1 mg/ml). The plates were incubated at 37 °C overnight. Single colonies were picked and propagated in 50 ml of LB medium supplemented with ampicillin (0.1 mg/ml) at 37 °C overnight. The bacterial pellet was obtained by centrifugation at 2880 × $g$ for 20 min. Plasmid DNA was extracted from the pellet using GeneJET Plasmid Miniprep Kit (Thermo Fisher, Catalog number K0503) according to the manufacturer's recommendations with modifications for larger volumes of starting bacterial culture. Plasmid DNA was eluted in 80 μL of nuclease-free water. DNA concentration was measured using Qubit™ dsDNA BR Assay Kit (Thermo Fisher Scientific, Catalog number Q32851) according to the manufacturer's recommendations.

### Experimental design
The circular DNA plasmid was digested or relaxed depending on the type of DNA template required. The plasmid was digested to obtain linear DNA using DraIII-HF or treated with *Escherichia coli* Topoisomerase I to only relax the circular plasmid. Once the plasmid was cut or relaxed, it was purified using a DNA purification kit. The purified DNA was then transcribed with T7RNAP. After transcription, the reaction was treated with DNase I to remove the DNA templates, and RNA was purified using an RNA purification kit. Finally, purified RNA was hybridized with DNA oligonucleotides to produce RNA ID, which were then characterized using nanopore sensing.

### Circular DNA plasmid digestion
DraIII-HF (New England Biolabs (NEB), R3510S) was used to linearize the circular DNA plasmid (sequence in Supplementary Table 1) following the manufacturer's recommendations. In a 50 μL reaction, 2000 ng of circular DNA plasmid were mixed with 2 μL of DraIII-HF (40 units), 5 μL of 10X rCutSmart Buffer, and nuclease-free water to achieve final reaction volume. The reaction components were mixed by pipetting and spin down for a couple of seconds. The reaction was incubated at 37 °C for 1 h. Enzymes were kept on ice throughout the whole preparation procedure. After DraIII-HF treatment, DNA was purified using Monarch PCR & DNA Cleanup Kit (5 μg) (NEB, T1030S).

ScaI-HF (20,000 units/mL, NEB, Catalog number R3122S) was used to linearize circular DNA plasmid for Supplementary Fig. 24 (plasmid sequence in Supplementary Table 5). A reaction of 2000 ng of DNA was prepared according to the manufacturer's recommendations. In a 50 μL reaction, 2000 ng of circular DNA plasmid were mixed with 2 μL of ScaI-HF (40 units), 5 μL of 10X rCutSmart Buffer, and nuclease-free water to reach 50 μL. The reaction was incubated at 37 °C for 1 h. DNA was also purified after digestion with ScaI-HF.

### Circular DNA plasmid relaxation
*Escherichia coli* Topoisomerase I (NEB, M0301) was used to relax a supercoiled circular DNA plasmid (sequence in Supplementary Table 1). A reaction of 2000 ng of DNA was prepared according to the manufacturer's recommendation. In a 100 μL reaction, 2000 ng of circular DNA plasmid were mixed with 3 μL of Topoisomerase I (15 units), 10 μL of 10X rCutSmart Buffer, and nuclease-free water to achieve the final reaction volume. The reaction components were mixed by pipetting and spin down for a couple of seconds. The reaction was incubated at 37 °C for 1 h, followed by incubation at 65 °C for 20 min to inactivate the enzyme. Enzymes were kept on ice throughout the whole preparation procedure. Topoisomerase I-treated DNA was purified using Monarch PCR & DNA Cleanup Kit (5 μg) (NEB, T1030S) following the manufacturer's instructions.

### DNA purification
Linearized or relaxed DNA was purified using Monarch PCR & DNA Cleanup Kit (5 μg, NEB, Catalog number T1030S). Binding and washing

of the sample to the purification columns were performed as suggested by the manufacturer, following the suggested centrifugation protocols. After washing, the DNA was eluted with preheated (50 °C) nuclease-free water. The elution step was performed twice with 10 μL of nuclease-free water after 5 min incubation at room temperature to increase DNA yield. The concentration of DNA was estimated using a NanoDrop spectrophotometer.

### T7 RNA polymerase in vitro transcription of DNA
Purified linear DNA and relaxed circular DNA were both in vitro transcribed using HiScribe™ T7 Quick High Yield RNA Synthesis Kit (NEB, Catalog number E2050S). For each, a 240 ng reaction was prepared according to the manufacturer's recommendation. In a 20 μL reaction, 240 ng of purified DNA were mixed with 2 μL of T7 RNA Polymerase (T7RNAP) Mix, 10 μL of NTP buffer mix (to achieve 10 mM concentration of each NTP), and nuclease-free water to achieve final reaction volume. The reaction components were mixed by pipetting and spin down for a couple of seconds. The reaction was incubated at 37 °C for 4 h.

After the 4 h of incubation (required for transcription), RNA was directly treated with 4 units of DNase I (NEB, M0303S) to remove linear or circular DNA used as a template for transcription. After DNA removal, RNA was purified using Monarch RNA Cleanup Kit (50 μg) (NEB, T2040S). Variability in the yield of RNA synthesis was observed between different lots of the same T7 RNA polymerase Mix. We attribute this to variability in the concentration of the active enzyme between lots. Yield could be adjusted by varying the concentration of T7 RNA polymerase Mix in the reaction mixture, and yield was consistent within lots. Enzymes were kept on ice throughout the whole preparation procedure.

### DNA removal: DNase I treatment
Transcription products were treated with DNase I (2000 units/mL, NEB, Catalog Number M0303S) to remove DNA templates. About 68 μL reaction mixture was prepared by adding 2 μL (4 units) of DNase I and 46 μL of nuclease-free water to the full volume (20 μL) of transcription products. The reaction components were mixed by pipetting and spin down for a couple of seconds. The reaction mixture was incubated at 37 °C for 15 min

### RNA purification
After DNA removal, RNA was purified using Monarch RNA Cleanup Kit (50 μg, New England Biolabs, Catalog Number T2040S) following the manufacturer's instructions for binding, washing and elution of the sample. RNA elution step was performed twice with 10 μL of nuclease-free water, allowing 5 min of incubation time per elution to maximize recovery.

### RNA identifier fabrication
To fabricate an RNA ID, purified RNA was mixed with complementary oligos, repeats label "R" oligos, and "1" bit oligos. A 40 μl reaction included 800 fmol of target RNA (or 20 nM in final volume) and 2400 fmol of the complementary oligos (or 60 nM in final volume). For the inclusion of repeats label "R" and "1" bits, 2400 fmol of the oligos containing the docking strand were added. The sequences of the oligos used for RNA ID assembly of transcripts produced from linear DNA templates are found in Supplementary Table 2 and oligos implemented for RNA ID assembly of transcripts produced from rolling circle transcription are found in Supplementary Table 3. The imaging strand was added in 1.5 times excess (10.8 pmol or 270 nM) to the three available docking strands; one from the repeats label "R" and two from the two "1" bits in the design. The mixture was done in 100 mM LiCl, 1 × TE buffer (10 mM Tris-HCl buffer, 1 mM ethylenediaminetetraacetic acid, pH 8.0), and nuclease-free water was added to achieve the final reaction volume[35]. LiCl was used to prevent magnesium

fragmentation[53] and nuclease-degradation of RNA that relies on magnesium ions[54]. Prior to use, the nuclease-free water was filtered with MF-Millipore membrane filters (0.22 μm pore size) and irradiated with UV light for 10 min. The reaction components were mixed by pipetting and spinning down briefly. Assembly of the components was done by heating to 70 °C for 30 s and gradually cooling over 45 min to room temperature (90 cycles of 30 s where temperature decreases by 0.5 °C in each cycle). RNA IDs were filtered twice in 100 kDa cut-off Amicon filters to remove excess oligos. 10 mM Tris-HCl (pH 8.0) with 0.5 mM $MgCl_2$ was used as a washing buffer for filtration. After purification, RNA IDs were kept at 4 °C before nanopore and agarose gel characterization.

## Agarose gel electrophoresis

RNA, DNA, and RNA ID samples were run on a 1% (w/v) agarose gel prepared in fresh 1 × TBE buffer, with 0.02% sodium hypochlorite using autoclaved Milli-Q water for both the gel and running buffer preparation. The samples were run with 1 × TriTrack loading dye (Thermo Fisher, Catalog number R1161). A constant voltage of 70 V was applied for 180 min, and 150 ng of the sample was added to each lane. The gel was stained in 3 × GelRed buffer (Biotium) and imaged with a GelDoc-It™(UVP). Gel images were processed using ImageJ (Fiji)[55]. The grayscale was inverted, the contrast was increased, the background was subtracted with a rolling ball of 50–150 and the surface was smoothened.

## Nanopore fabrication

Glass nanopores were produced from quartz glass capillaries with 0.5 mm outer diameter and 0.2 mm inner diameter (Sutter Instruments, USA) using a laser-assisted capillary puller P-2000 (Sutter Instruments, USA). The nanopores had diameters of 8 to 15 nm[56], achieved by using the following heating parameters of the instrument: HEAT, 470; VEL, 25; DEL, 170; and PUL, 200.

## Microfluidic chip fabrication for nanopore measurements

A polydimethylsiloxane (PDMS) microfluidic chip was fabricated for nanopore measurements. The chip contained small chambers arranged in a radial geometry with respect to a central chamber, connected to the rest through small channels. The chip was produced by combining a 10:1 volume ratio of PDMS monomer and curing agent (Sylgard 184 silicone elastomer kit, The Dow Chemical Company), which were mixed for 10 min and then poured into a preheated mold (60 °C, 3 min). The mold was put into a vacuum to remove air bubbles in PDMS and then it was heated at 60 °C for 48 h. The PDMS chips were removed from the mold and small perpendicular holes were created in the channels connecting the chambers. Then, glass nanopores were placed in the channels and the chip was sealed by pressing it against a glass slide after being treated for 11 s in a plasma chamber using maximum power (Femto, Diener). The channels were then sealed by introducing the PDMS mixture (10:1) in the perpendicular orifices and the chip was heated in a hot plate at 150 °C for 5 min, followed by 100 °C heating for 60 min. After cooling down, the chip was treated for 5 min in a plasma chamber at maximum power and all the chambers were filled with 4 M LiCl.

## Nanopore measurements

For nanopore measurements, RNA ID was diluted to 400 pM in 4 M LiCl, 1 × TE, pH 9.4. A fixed potential of 600 mV was used for all measurements, and data was recorded using an Axopatch 200B amplifier (Molecular Devices) and filtered with the 100 kHz internal filter of the Axopatch amplifier. An eight-pole analog low-pass Bessel filter (900CT, Frequency Devices) with a cut-off frequency of 50 kHz was also used. The data were acquired with a data card (PCI-6251, National Instruments) using a sampling frequency of 1 MHz. The IV curves for the nanopores presented in this work are presented in Supplementary Table 8.

Single translocation events were isolated from the raw ionic current trace using event charge deficit (area of the event), mean current, and translocation time, using home-built LabVIEW codes. Event charge deficit boundaries were set from 0 to 400 fC. The translocation time minimal threshold was set to 0.05 ms and the minimal mean current drop threshold was −100 pA. This enabled us to visualize events from RNA ID translocations while filtering out random RNA blobs or loose streptavidin. For further study of the RNA IDs, unfolded RNA ID (linear RNA ID) was selected based on the ionic current trace of individual events. We identified the number of downward current spikes in the events and used the distance between spikes to associate them with our initial designs and categorize them accordingly. Supplementary Fig. 27 shows the ratio of total events and compares it with the events that were sized. Supplementary Fig. 28 includes further details on the data analysis performed, and Supplementary Fig. 29 demonstrate that the distinctive current traces characterized emanate exclusively from RNA ID translocation.

## Single-molecule sizing

A Python-based graphic user interface was used for single-molecule sizing of RNA ID. In each translocation event, downward current spikes associated to components of RNA ID design with known base pair distance were selected manually. The known base pair separation between the spikes was associated with the time distance between both spikes, to obtain a base pair-to-time conversion factor. The start and end points of each event were selected manually, and the total event translocation time was converted to an estimate of base pairs using the respective conversion factor for each RNA ID translocation.

## Materials

Commercial reagents implemented in this work included nuclease-free water (Ambion, catalog number AM9937), 100 × Tris-EDTA buffer solution concentrate (Sigma-Aldrich, Catalog number T9285), Lithium chloride for molecular biology ≥99% purity (Sigma-Aldrich, Catalog number L9650), Tris-HCl BioPerformance certified, ≥99% purity (Sigma-Aldrich, catalog number T5941). Buffer solutions prepared from these reagents were filtered with 0.22 μm Millipore syringe filter units (MF-Merck Millipore™, Catalog number GSWP04700).

For enzymatic reactions the following products were used: DraIII-HF (20,000 units/mL, New England Biolabs (NEB), Catalog number R3510S), ScaI-HF (20,000 units/mL, NEB, Catalog number R3122S), *Escherichia coli* Topoisomerase I (5000 units/mL, NEB, Catalog Number M0301S), DNase I (2000 units/mL, NEB, Catalog Number M0303S), HiScribe™ T7 Quick High Yield RNA Synthesis Kit (NEB, Catalog number E2050S).

DNA and RNA purification were performed with the following purification kits: Monarch PCR & DNA Cleanup Kit (5 μg, NEB, Catalog number T1030S) for DNA, and Monarch RNA Cleanup Kit (50 μg, New England Biolabs, Catalog Number T2040S) for RNA.

For this study we used Labcon Eclipse™ 10 μL Graduated Pipette Tips with UltraFine™ Point (Thermo Fisher Scientific, Catalog number 16603912), Eppendorf DNA LoBind® Tubes (Thermo Fisher Scientific, Catalog number 10686313 and 107008704), 0.2 mL RNase-free PCR tubes (Thermo Fisher Scientific, Catalog number AM12225), and Invitrogen RNaseZap™ RNase Decontamination Solution (Thermo Fisher Scientific, Catalog number AM9780).

For the fabrication of glass nanopores, we used glass quartz capillaries with 0.2 mm inner diameter and 0.5 outer diameter (Sutter Instrument Company). For chip fabrication, we used Sylgard 184 PDMS (Dow Corning, Catalog number 101697), microscope slides clear ground 1.0–1.2 mm (Thermo Fisher Scientific, Catalog number 1238-3118).

## Statistics and reproducibility

Each nanopore experiment was continuously run to collect as many RNA ID translocations. Data acquisition was typically limited to blocking of the pore. For this reason, there is no set sample size, however, we only use experiments where we measured at least 100 unfolded RNA IDs. In our measurements, we select unfolded RNA ID translocations for further analysis, which allows us to assign RNA IDs more precisely. The experiments presented in this work were performed in at least three different nanopores. All replication attempts were successful. The experiments were not randomized. The investigators were not blinded to allocation during experiments and outcome assessment.

## Reporting summary

Further information on research design is available in the Nature Portfolio Reporting Summary linked to this article.

## Data availability

Data supporting the findings of this study are available in the main text and the Supplementary Materials. Raw current traces are provided in Supplementary Materials, and source data is provided with this paper. Additional raw data are available at https://doi.org/10.17863/CAM. 104528. Source data are provided with this paper.

## Code availability

The data presented was analysed using home-built software written with National Instruments LabVIEW 2017 and Python. All the data presented in this work and in the Supplementary Materials can be plotted and analysed manually or using any software of preference.

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

## Acknowledgements

U.F.K. was supported by funding from a European Research Council (ERC) consolidator grant (DesignerPores no. 647144) and an ERC-2019-PoC grant (PoreDetect no. 899538). G.P.-G. acknowledges funding from EPSRC CDT MRes/PhD Studentship in Nanoscience and Nanotechnology (NanoDTC Cambridge EP/S022953/1) and Trinity-Henry Barlow Scholarship. F.B. acknowledges research funding from the George and Lilian Schiff Foundation Studentship, the Winton Program for the Physics of Sustainability PhD Scholarship, and St John's College Benefactors' Scholarship. J.P., M.P., and D.S.-P. acknowledge funding from the Science Fund of the Republic of Serbia, Grant No. 7754217, READ-DM1. We thank Dr. Casey Platnich and Dr. Roger Rubio-Sánchez for critically reading the manuscript. We thank Lorenzo Peri for facilitating python-based graphic user interfaces used for analysis of data.

## Author contributions

G.P.-G., F.B., U.F.K., and J.P. designed the study. G.P.-G. designed and performed all nanopore experiments and analysed the data. J.P. and G.P.-G. designed and performed agarose gel electrophoresis. DNA plasmids were designed, propagated, and isolated by M.P., J.P., and D.S.P. Finally, G.P.G., F.B., and U.F.K. wrote the manuscript with input from J.P., M.P., and D.S.-P., F.B., and U.F.K. supervised the study.

## Competing interests

F.B. and U.F.K. are inventors of two patents related to RNA analysis with nanopores (UK patent application no. 2113935.7, in process; UK Patent application nos. 2112088.6 and PCT/GB2022/052171, in process) submitted by Cambridge Enterprise on behalf of the University of Cambridge. U.F.K. is a co-founder of Cambridge Nucleomics. The remaining authors declare no competing interests.
