## [Peer Review File · Nature Communications]

REVIEWER COMMENTS

Reviewer #1 (Remarks to the Author):

The manuscript endeavors to craft a quantitative method to elucidate alternative transcription termination alongside the processivity of T7 RNA polymerase. This is achieved by engineering RNA identifiers and harnessing nanopore sensing. Their methodology involved transcription of a linear DNA equipped with a T7RNAP promotor sequence and devoid of a terminator sequence. Subsequent to this, each RNA product underwent hybridization with cDNA oligos, culminating in the formation of an RNA-DNA duplex or the RNA identifier (RNA ID). They further harnessed a circular DNA construct devoid of the termination sequence to scrutinize the rolling circle transcription of T7RNAP. Based on the transcripts from both linear and circular DNA, the authors identified distinct transcripts and depicted T7RNAP using parallel single-transcript mapping via nanopore sensors. While this research extends the repertoire of methods for transcript analysis, I have identified several pivotal issues that the authors must address to enhance the quality and rigor of the manuscript.

Major comments:

The manuscript suggests that sequence information of the transcripts' template is essential for designing the RNA ID (ss DNA oligonucleotides). Is there an alternative strategy for designing RNA ID for transcripts with undisclosed sequence information?

Within Supplementary Figure 3, the evidence indicates a nearly twofold variation in translocation time for identical PT RNA IDs. How do the authors account for this disparity? Moreover, certain translocation events depict negligible variance in current peaks between label 'R' and label '1-bit'. Can the authors elucidate this occurrence?

To ascertain that the captured current signals emanate exclusively from the intended transcripts, and not contaminants or other solution constituents, the manuscript should incorporate the figure detailing the current-time domain in the absence of analytes in the nanopores.

Supplementary Information, Figure 5c & d, reveals overlapping regions among the three distributions. Can the authors devise a methodology to delineate these clusters more distinctly? Exploring voltage variations beyond 600 mV might be advantageous.

In Supplementary Information, Figure 18a, a conspicuous charge deficit distribution exists between the yellow and green dash lines. Why haven't the authors acknowledged this as a discrete conformation?

Minor comments:

Within the supplementary information segment titled “Single-molecule sizing”, manual selection of event endpoints is mentioned. Leveraging scripts based on defined thresholds might proffer more consistent and precise determination of event boundaries.

Supplementary Information, Figure 16, for $n=5$, depicts a singular translocation signal for each nanopore size. For robust repeatability validation, the authors should incorporate three replicates for each nanopore dimension.

In Supplementary Information, Figure 16, for $n=6, 7, 8$, the translocation signals remain uncolored. The authors must assign the appropriate colors (either red or blue) for enhanced clarity.

Essential metrics, including the sampling rate, low pass Bessel filter parameters, and raw data processing thresholds, should be provided. This will affirm the validity of the data collection and subsequent processing.

Several figures within the supplementary information possess inconsistencies. Specifically, Figures 5a and 5b exhibit varied rectangle dimensions and distinct y-axis range capabilities. These elements should be standardized, particularly for contrasting DNase I treatment outcomes.

Reviewer #2 (Remarks to the Author):

In this article, the authors use glass pipette nanopores to study the transcription process of a design DNA sequence to analyze the potential termination efficiency of a known sequence (OriC). This work relies on a large amount of biochemical experiments and nanopore experiment. It is overall totally convincing, and the data shown supports the main scientific question raised in the introduction. Nevertheless, some points could be detailed or discuss to enrich the article. I thus recommend the publication of this article after the authors addressed the major and minor following remarks :

Major comments

- The author implicitly claimed (line 138 to 140) that, unlike gel electrophoresis, their method could give information about the sequence of the analyzed RNA. But all the presented results rely on a known RNA sequence. The bit ‘1’ and ‘R’ oligos which are tagging the probed molecules have a specific sequence that is specific to the designed plasmid of this study. The author should rephrase or comment on this point. The same remark can be made about the claim of the first paragraph on the discussion section.
- The interesting point of stochastic dissociation of the RNAP is mentioned first on line 282 of the manuscript in the introduction of the rolling cycle experiment. What about these dissociations in the first experiment ? Is it possible to see them to quantify them ? Are they included in the distribution width ? The author should comment on this.

- It seems that some RNAP processivity characterization could be achieved with this set of experiment. For instance, isn't it possible to extract some kinetics information from the statistics of the number 'n' of cycles of the transcription (FIGURE 17A) ? This is absolutely not discussed in the paper but is of interest.
- Around line 390, the authors provide information about different DNA constructs with different length between the RNAP promoter and the OriC. The corresponding nanopore experiments would be valuable for the article.
- The proportion of "good" events (unfolded and showing the 'R' marker and the '1' bits) compared to the total number of events recorded should be provided for a sake of transparency about the difficulty of the experiment. This would avoid vague expression about the representativity of the display event (line 368)

Minor comments

- On line 146, "which revealed" should be replaced by "which confirmed" as the two types of RNA were revealed earlier in the text using gel electrophoresis.
- On line 165, in the sentence "Besides, the rest of the RNA is decorated with '1' bits" should be rephrased. It is indeed misleading (only the beginning of the RNA is decorated. The END event on figure 1e shows exactly that there is no decoration on the second half of the RNA.
- The translation of translocation time to RNA length assumes that the speed is constant along one translocation. In the $n > 2$ experiments, the time between 'R' markers should provide some insight about this.
- On line 345 about figure 3e, the plot does not show a linear dependence with 'n' but rather a linear dependence of the charge deficit with the translocation time. These informations are provided in figure 17b (and are in my opinion, more interesting than the figure 3e linear dependency).
- The pore size should be provided for all data presented.
- In supplementary figure 5, there is an inconsistency of the total number of events in the figure and its legend. Figure 5a says 1500 events and the legend says 1700 events.
- Add the proportion of 'unfolded event'
- In Supplementary figure 5b the red and grey distribution seems to be the one presented on figure 2c. Is this correct ? please clarify.
- We could expect to see an equal number of 3'->5' and 5'->3' events. But the 'representative' (line 368) events in the supplementary figure. For instance the SFigure 12 shows only 5'->3' events... is this really representative ? Do the author have an explanation of this if yes.
- SFigure 18, is there a specific meaning to invert the order of the plots in 18a part compare to 18b part ?

- The verb 'to enable' is used about 30 times throughout the main text and especially 4 times between line 75 and line 90. The author should consider synonyms to avoid reading's heaviness.

Reviewer #3 (Remarks to the Author):

What are the noteworthy results?

The paper showcases the application of the technology of RNA-ID developed at the Keyser lab to studying in vitro RNA transcription with T7 Polymerase. In the paper, authors encounter premature transcription termination that happens at OriC with around 50% probability. The authors use RNA ID technology in conjunction with nanopore readout, offering a more cost-effective alternative to traditional methods like RNE-seq. This approach also overcomes the challenges associated with performing long RNA reads.

Will the work be of significance to the field and related fields? How does it compare to the established literature? If the work is not original, please provide relevant references.

The work is original, and the fact of premature transcription termination is both unexplored and very intriguing. The work definitely represents an asset for in vitro studies of DNA transcription by various proteins and their modifications. The significance of the work spreads into the field of microbiology, biophysics, single-molecule biophysics, and also shows great utility for the nanopore sensing platform.

Does the work support the conclusions and claims, or is additional evidence needed?

The claim that nanopore technology combined with RNA ID tagging can be used to study transcription is fully supported by the work. The claim of studying the T7 polymerase transcription prompts some questions, that could be addressed in this cycle of revision:

1) Is only the OriC terminating the transcription prematurely, or do other Origins trigger that? (I think if experiments are needed to answer this question, then they don't belong not the scope of this paper, but maybe indicated in text)

I am missing a discussion. Why, if OriC terminates the transcription, does the entire transcript of OriC end up in the RNA? 2) Is there a part of the OriC that acts as a terminator, or is it the entire sequence?

(this question I think can be well addressed with the methodology developed, or at least the prospect of it should be indicated in text. I don't think extra experiments are needed to publish this paper).

3) Figure 3d contains a gel where in the first lane we have RNA product of circular rolling transcription and the lines in it indicate that it's most often terminated in a certain spot all the time. But I see faint lines at around 3kb (between $n=2$ and $n=1$) and maybe (not sure because of the color boxes) between $n=2$ and $n=3$. It would be good to comment on those, and if those are true RNA products, I would expect that there should be events, which don't only end with R11 ID, but extend further.

4) This question stems more or less from the previous one: Were there any prematurely terminated events not at the origin? There should be some probability of it happening.

5) The authors derive the probability of dissociation from the OriC based on first experiments with linear DNA, but I think based on the rolling transcription it is possible to calculate the entire binomial statistics and also deduce this probability. It would be good to compare both.

Are there any flaws in the data analysis, interpretation, and conclusions? Do these prohibit publication or require revision?

I don't think there are any flaws in data analysis/conclusions, but the discussion could be stronger based on my previous questions. This manuscript would require revision based on the questions above.

Is the methodology sound? Does the work meet the expected standards in your field?

Yes, I praise the methodology, cleanliness of the event signatures and transparency of the manuscript. This paper shows definitely one of the most state-of-the-art nanopore experiments.

Is there enough detail provided in the methods for the work to be reproduced?

The biological part is pretty clear, and the nanopore part refers to the necessary literature that describes the nanopore methodology in greater detail. Nanopore technology is in general not easy to reproduce by non-nanopore lab, but I think based on this paper, at least another nanopore lab can repeat these experiments.

PS I have attached a pdf with some of my questions marked. They may overlap with the ones presented here, but also could be minor suggestions for the text. (For example, the motivation paragraph could be written out more strongly)

Response letter to the reviewers for the manuscript:

Single-Molecule RNA Sizing Enables Quantitative Analysis of Alternative Transcription Termination

Patiño-Guillén¹, G., Pesovic, J.², Panic, M.^{2,3}, Savic-Pavicevic, D.², Bošković, F.^{1,*}, Keyser, U.F.^{1,*}

¹Cavendish Laboratory, University of Cambridge, Cambridge, UK

²University of Belgrade – Faculty of Biology, Centre for Human Molecular Genetics, Belgrade, Serbia

³Institute of Virology, Vaccines and Sera "Torlak", Belgrade, Serbia

*Corresponding authors: fnb24@cam.ac.uk , ufk20@cam.ac.uk

We are grateful to the reviewers for the time devoted to assessing our manuscript. In addition, we appreciate the reviewers' valuable comments and corrections. Below, our answers to each comment and correction suggested are listed. We have adapted the manuscript according to the reviewers' requests. The reviewers' points are in italic and our responses are in regular font. The modifications implemented in the manuscript are highlighted in red. Some of the comments are of similar substance, hence we helped the reviewers to follow the changes by repeating the section that is required by both reviewers.

Reviewer #1 (Remarks to the Author): *The manuscript endeavors to craft a quantitative method to elucidate alternative transcription termination alongside the processivity of T7 RNA polymerase. This is achieved by engineering RNA identifiers and harnessing nanopore sensing. Their methodology involved transcription of a linear DNA equipped with a T7RNAP promotor sequence and devoid of a terminator sequence. Subsequent to this, each RNA product underwent hybridization with cDNA oligos, culminating in the formation of an RNA-DNA duplex or the RNA identifier (RNA ID). They further harnessed a circular DNA construct devoid of the termination sequence to scrutinize the rolling circle transcription of T7RNAP. Based on the transcripts from both linear and circular DNA, the authors identified distinct transcripts and depicted T7RNAP using parallel single-transcript mapping via nanopore sensors. While this research extends the repertoire of methods for transcript analysis, I have identified several pivotal issues that the authors must address to enhance the quality and rigor of the manuscript.*

Thank you for taking the time to review our manuscript and for your comments. We appreciate your recognition of our efforts to expand in a unique way single transcript analysis with nanopore sensors.

Below you can find our replies and changes implemented to address the issues you mentioned.

Major comments: The manuscript suggests that sequence information of the transcripts' template is essential for designing the RNA ID (ss DNA oligonucleotides). Is there an alternative strategy for designing RNA ID for transcripts with undisclosed sequence information?

We agree with the reviewer that it is an interesting question if we can detect unknown RNA sequences. It is likely possible to design a process inspired by sequencing-by-hybridisation to analyse unknown RNA sequences. In this work we concentrate on known sequences which is a major strength of our technique. The technique allows us to identify which parts of the gene's sequence is retained on the transcript level. In future, we may revisit unknown sequence detection, but we have clarified the text to stress that we concentrate on known sequences.

We have clarified that we look at known sequences in the main text following the referee's point (page 4, line 145):

'Our technique allows the identification of transcript variants resulting from alternative transcript processing solely based on the DNA sequence of the gene.'

Within Supplementary Figure 3, the evidence indicates a nearly twofold variation in translocation time for identical PT RNA IDs. How do the authors account for this disparity?

The factor of two in translocation time is a direct consequence of the free translocation through our nanopores. We have quantified the variability in translocation time in prior research (Chen et al, Nature Physics, 2021). The physical configuration of a molecule, specifically RNA ID, influences the translocation time variability due to the inherent fluctuation of the polymer coil.

We added a statement to address the referee's point in the supplementary figure 3, indicated by the reviewer (page 8):

'Nanopore event variability can be accounted to the physical configuration-dependent RNA ID transport²'.

Moreover, certain translocation events depict negligible variance in current peaks between label 'R' and label '1-bit'. Can the authors elucidate this occurrence?

We agree that – like the translocation time – the spike depth in some events presented in Figure 3 is variable. Detailed description of monovalent streptavidin contribution to spike depth in ionic current readout has been previously performed (Boskovic, F. and Keyser U.F., Nature Chemistry, 2022). The depth of the spike depends on the measurement bandwidth, translocation time, and the conformation of the molecule in the nanopore. We would like to emphasize that the mean current spike depth of labels with two streptavidins is larger in magnitude than single streptavidin. To identify the events, we use the spike depth and position relative to markers guided by the known design of our molecules. As

we have shown in the Nature Chemistry 2022, we can add more streptavidin binding sites per label to increase the difference between both types of labels and hence aid identification if necessary.

To ascertain that the captured current signals emanate exclusively from the intended transcripts, and not contaminants or other solution constituents, the manuscript should incorporate the figure detailing the current-time domain in the absence of analytes in the nanopores.

Our RNA IDs enable an additional level of specificity. Some previous papers have demonstrated that a chance for mislabelling our IDs with multiple labels is negligible. (Boskovic, F. and Keyser U.F., Nature Chemistry, 2022, and Boskovic et al, Nature Nanotechnology, 2023). However, to address the reviewer's concern, we have performed a nanopore measurement where the RNA ID was not included to show the effect of potential background constituents in the current trace. This is shown in new Supplementary Figure 29:

We have also included the following statement in the manuscript (page 5, lane 187):

'The position of the labels in the RNA ID produces a characteristic ionic current readout, which provides an additional level of specificity to exclude false-positive detection. Hence, we can exclude that contaminants create such events as demonstrated previously^{34,35}.'

Figure 29

Figure 29: Characterization of ionic current trace shows that distinctive current traces emanate exclusively from the RNA IDs. **a** A nanopore measurement was performed in the absence of RNA IDs for 90 minutes. Translocation events detected using the following threshold parameters are shown. Minimum charge deficit: 0 fC, maximum charge deficit: 400 fC, minimum translocation time: 50 μs, minimum current drop: -100 pA. Ionic current scale bar corresponds to 500 pA, and translocation time scale bar to 200 μs. **b** The charge deficit of the translocations detected are plotted, demonstrating that the current traces studied in this work originate from the translocation of RNA IDs.

Supplementary Information, Figure 5c & d, reveals overlapping regions among the three distributions. Can the authors devise a methodology to delineate these clusters more distinctly? Exploring voltage variations beyond 600 mV might be advantageous.

The plots of the mean event current versus translocation time (Supplementary Figure 5c and 5d) only represent a bulk plot of all data points prior to nanopore event analysis. We have included an additional section in the Supplementary Information (Supplementary Figure 28) to provide further detail on the way the data is analysed at single molecule level:

Figure 28

Figure 28: Step-by-step nanopore data analysis for RNA ID sizing. **a** An exemplary region of the ionic raw current trace is shown. We use a home-built LabVIEW code to identify translocation events by simple thresholding. Events are then analysed by using standard parameters, which include the charge deficit, minimum translocation time and mean current that are given by the length of the RNA ID molecule and its design. **b** The translocation events were then categorized into PT RNA ID, END RNA ID, folded RNA ID and translocations associated to misfolded molecules (marked X) based on the ionic current trace of the events. **c** The base pair length of PT RNA IDs and END RNA IDs was computed using the RNA ID design. The distance between current spikes (blue) was measured (in time units). This distance corresponds to a known base pair length which was used to calculate a conversion factor. Then, the length in time units of the entire translocation event was measured (pink) and converted into a base. The base pair length was plotted as shown in the right.

In Supplementary Information, Figure 18a, a conspicuous charge deficit distribution exists between the yellow and green dash lines. Why haven't the authors acknowledged this as a discrete conformation?

We agree with the reviewer that events in Fig. 18a are distributed between the dashed lines. As discussed above, the translocation time of molecules even with the same length may vary by a factor of two due to molecular conformations. However, the associated structures and colours can still be assigned with high confidence by checking individual nanopore events of the RNA IDs. We believe that the events between yellow and green markers mainly stem from dissociation of RNA polymerase, between two pinpointed cases. The main reason for this interpretation is that the distribution does correspond to RNA transcripts as it is in agreement with agarose gels presented in Figure 3 and supplementary Figure 10.

We provided a description to assure that this is clear to the reader (Supplementary material, Figure 18):

'The charge density distributions produced between the main distributions (green and yellow) are ascribed to fall-off of T7RNAP.'

Minor comments:

Within the supplementary information segment titled "Single-molecule sizing", manual selection of event endpoints is mentioned. Leveraging scripts based on defined thresholds might proffer more consistent and precise determination of event boundaries.

After revising the comments from all reviewers, we realized that a more detailed explanation of nanopore data analysis was required to clarify these concerns. We have now included a figure in the supplementary materials, Supplementary Figure 28 that explains in detail how raw data is selected, events categorized and the sizing performed. We have also explained in more detail nanopore data analysis in the methods section (page 21, line 551):

'Single translocation events were isolated from the **raw** ionic current trace using **event** charge deficit (area of the event), mean current, and translocation time, using home-built LabVIEW codes. **Event charge deficit boundaries were set from 0 to 400 fC. The translocation time minimal threshold was set to 0.05 ms and the minimal mean current drop threshold was -100 pA.** This enabled us to visualize events from RNA ID translocations while filtering out random RNA blobs or loose streptavidin. For further study of the RNA IDs, unfolded RNA ID (linear RNA ID) was selected **based on the ionic current trace of individual events.'**

Supplementary Information, Figure 16, for $n=5$, depicts a singular translocation signal for each nanopore size. For robust repeatability validation, the authors should incorporate three replicates for each nanopore dimension.

In Supplementary Figure 16 we included the additional events:

Figure 16

Figure 16. RNA ID nanopore events for transcription cycles $N = 5$ and $N > 5$ example events. **a** Example of nanopore events of RNA IDs produced from five transcription cycles ($N = 5$) measured in pores with different sizes. All measurements were performed under the same applied voltage of 600 mV. **b** Also, RNA IDs with $N > 5$ were identified, example events are shown from different nanopore measurements. **c** Translocation of RNA IDs for each transcription cycle that entered to the pore in the 5' to 3' direction.

In Supplementary Information, Figure 16, for $n=6, 7, 8$, the translocation signals remain uncolored. The authors must assign the appropriate colors (either red or blue) for enhanced clarity.

We coloured the events of Supplementary Figure 16b for consistency as suggested.

Figure 16

Figure 16. RNA ID nanopore events for transcription cycles $N = 5$ and $N > 5$ example events. **a** Example of nanopore events of RNA IDs produced from five transcription cycles ($N = 5$) measured in pores with different sizes. All measurements were performed under the same applied voltage of 600 mV. **b** Also, RNA IDs with $N > 5$ were identified, example events are shown from different nanopore measurements. **c** Translocation of RNA IDs for each transcription cycle that entered to the pore in the 5' to 3' direction.

Essential metrics, including the sampling rate, low pass Bessel filter parameters, and raw data processing thresholds, should be provided. This will affirm the validity of the data collection and subsequent processing.

Sampling rate, low pass Bessel filter parameters, and raw data processing thresholds have now been included in the methods section of the manuscript (page 20, line 545). We have also included IV curves for the nanopore experiments performed, this is now presented in Supplementary Table 8, below.

'A fixed potential of 600 mV was used for all measurements and data was recorded using an Axopatch 200B amplifier (Molecular Devices) and filtered with the 100 kHz internal filter of the Axopatch amplifier. An eight-pole analogue low-pass Bessel filter (900CT, Frequency Devices) with a cut-off frequency of 50 kHz was also used. The data was acquired with a data card (PCI-6251, National Instruments) using a sampling frequency of 1MHz. The IV curves for the nanopores presented in this work are presented in Supplementary Table 8.

Single translocation events were isolated from the raw ionic current trace using event charge deficit (area of the event), mean current, and translocation time, using home-built LabVIEW codes. Event charge deficit boundaries were set from 0 to 400 fC. The translocation time minimal threshold was set to 0.05 ms and the minimal mean current drop threshold was -100 pA. '

Table 8.

The table shows the IV curves of nanopores used to study RNA IDs. The ionic current and RMS noise at 600 mV is presented for each pore. An estimate of each nanopore diameter was calculated from the ionic current values presented, assuming a conical pore geometry⁵.

Table 8

Pore number	Sample	Current, RMS noise at 600 mV and calculated pore diameter	Current / voltage curve (IV curve)
-------------	--------	---	------------------------------------

1 RNA IDs produced from linear DNA
11.4 nA
6.5 pA
~8 nm

2 RNA IDs produced from linear DNA, and DNA template
9.9 nA
6.7 pA
~7 nm

3 RNA IDs of produced from circular template
8.7 nA
6.2 pA
~6 nm

4

RNA IDs
of produced
from circular
template

7.5 nA
6.6 pA
~5 nm

5

RNA IDs
of produced
from circular
template

12.2 nA
6.4 pA
~9 nm

6

RNA IDs
of produced
from circular
template

10.0 nA
6.4 pA
~7 nm

7

DNA ladder
SF8
(0.5 kbp –
10 kbp)

11.1 nA
5.7 pA
~8 nm

8

RNA IDs
produced
from 4.5
kbp DNA

10.9 nA
6.4 pA
~8 nm

An estimate the pore's diameter was calculated from the overall resistance of the nanopore in open state, R , which is constituted by the resistance of the pore cavity region, R_{pore} , and the resistance of the access region of the pore, R_{acc} ⁵:

$$R = R_{pore} + R_{acc}$$

This equation can be rewritten in terms of the resistivity ρ of the electrolytic solution, the pore's length, L , and the diameter of the *cis* and *trans* aperture of the pore, D_{cis} and D_{trans} :

$$R = \rho \frac{4L}{\pi D_{trans} D_{cis}} + \rho \left(\frac{1}{2 D_{trans}} + \frac{1}{2 D_{cis}} \right)$$

The diameter of the pore D_{cis} was calculated using the experimental ionic current I during the application of a 600 mV potential, assuming a D_{trans} of 200 μm , conductivity of 15.5 Sm^{-1} for 4M LiCl, and length L of 950 μm for our glass nanopores^{6,7}.

Several figures within the supplementary information possess inconsistencies. Specifically, Figures 5a and 5b exhibit varied rectangle dimensions and distinct y-axis range capabilities. These elements should be standardized, particularly for contrasting DNase I treatment outcomes.

We have corrected the format and modified axis range of this figure.

Figure 5

Figure 5. Characterization of RNA ID using charge deficit, mean current and translocation time. **a** Histogram of the charge deficit of RNA ID still in the presence of the linear DNA template. Histogram shows 3 distributions, the one with the lowest charge deficit is ascribed to premature termination (PT), the middle distribution corresponds to transcription of the full linear DNA (END) and the distribution furthest to the right is ascribed to the linear DNA template. This distribution (composed of 1500 events) includes translocations of molecules with multiple conformations, which include folded events, constructs with knots and unfolded events. **b** After treatment with DNase I, it can be seen how the distribution furthest to the right, ascribed to DNA, is removed. Unfolded translocations of both PT (red) and END (gray) RNA IDs (presented in Figure 2c) describe the entire sample, despite their conformation, while enabling single-molecule sizing. These distributions correspond to the same nanopore measurement presented in Figure 2. **c** Scatter plot of mean current against translocation time shows three distinct distributions, attributed to PT RNA IDs, END RNA IDs, and the linear template (from left to right). Unfolded molecules take longer to translocate through the pore than folded molecules but cover less cross-sectional area of the pore while translocating, producing a less significant drop in ionic current for longer times. Plotting events with different conformations produces this type of distribution, with unfolded events at the top, and

folded events at the bottom of each distribution. **d** Selection of unfolded events (in the sample treated with DNase I) is performed to describe each distribution. Unfolded events were found at the top of each distribution, and they are representative of the whole sample.

Reviewer #2 (Remarks to the Author): *In this article, the authors use glass pipette nanopores to study the transcription process of a design DNA sequence to analyze the potential termination efficiency of a known sequence (OriC). This work relies on a large amount of biochemical experiments and nanopore experiment. It is overall totally convincing, and the data shown supports the main scientific question raised in the introduction. Nevertheless, some points could be detailed or discuss to enrich the article. I thus recommend the publication of this article after the authors addressed the major and minor following remarks:*

We appreciate the reviewer's detail assessment of our manuscript and for acknowledging that our study relies 'on a large amount of biochemical experiments and nanopore experiment'. Below, we replied to the reviewer's points with the exact corrections in the main text that we implemented.

Major comments:

The author implicitly claimed (line 138 to 140) that, unlike gel electrophoresis, their method could give information about the sequence of the analysed RNA. But all the presented results rely on a known RNA sequence. The bit '1' and 'R' oligos which are tagging the probed molecules have a specific sequence that is specific to the designed plasmid of this study. The author should rephrase or comment on this point. The same remark can be made about the claim of the first paragraph on the discussion section.

We agree with the reviewer that this was unclear. To address the reviewer's comment, we included a sentence to explain that our method can be used to check for alternative transcript variants of a gene of interest if the DNA target sequence is known (page 4, line 145).

'Our technique allows the identification of transcript variants resulting from alternative transcript processing solely based on the DNA sequence of the gene.'

Moreover, we have also included a small discussion about this in the main text as suggested by the reviewer (page 17, line 429):

RNA IDs can be engineered to target genes of interest. Hence, we can produce labels to identify which regions of the gene's sequence is retained at transcript level.

The interesting point of stochastic dissociation of the RNAP is mentioned first on line 282 of the manuscript in the introduction of the rolling cycle experiment. What about these dissociations in the first experiment? Is it possible to see them to quantify them? Are they included in the distribution width? The author should comment on this.

It is possible for this to happen but rarely without a defined terminator. We have included this statement in the manuscript on the page 3, line 126 to address this point:

'T7RNAP was engineered to be a processive enzyme, hence we expect low dissociation without a defined terminator sequence⁸.'

In seems that some RNAP processivity characterization could be achieved with this set of experiment. For instance, isn't it possible to extract some kinetics information from the statistics of the number 'n' of cycles of the transcription (FIGURE 17A) ? This is absolutely not discussed in the paper but is of interest.

We fully agree with the referee that it would indeed be possible to gain kinetics information from T7RNAP with our experimental design. We have included a brief statement in the discussion section (page 18, line 459), indicating this possibility to the reader.

'RNA ID assembly coupled with nanopore sensing characterization can provide valuable insights into the kinetics of RNAPs.'

To describe processivity and termination in our rolling circle transcription system in a more quantitative way, decay functions were derived to describe premature termination at OriC and T7RNAP dissociation at a different position from the premature termination sequence. The decay functions were fitted from bulk nanopore measurement data. A new supplementary figure (Supplementary Figure 26) was added:

Figure 26

Figure 26: Quantitative description of T7RNAP processivity and transcription termination. **a** Considering a probability p of transcription terminating at the identified premature transcription termination site, and a probability p_s for transcription terminating solely by

dissociation of the polymerase at a different region of the plasmid. Equation $f(x)$ describes the abundance of transcripts originated from premature transcription termination at OriC. Equation $g(x)$ describes the transcript abundance originated from fall-off of T7RNAP at a different region of the plasmid. **b** The distribution of the charge deficit of nanopore translocation events is presented. The different transcript populations were fitted to gaussian functions, from which the relative abundance of transcripts was derived. The distributions ascribed to premature termination transcripts are labelled with x values of odd integers ($x = 1, 3, 5, 7$) and the minor distributions ascribed to transcripts produced from T7RNAP dissociation in a different region of the plasmid receive x values of even integers ($x = 2, 4, 6, 8$). **c** The relative abundance of transcripts (black) are plotted. Fitting of $f(x)$ and $g(x)$ is plotted in red, $p \sim 0.51$ and $p_s \sim 0.21$, which agrees with transcription termination reported in linear DNA constructs.

Around line 390, the authors provide information about different DNA constructs with different length between the RNAP promoter and the OriC. The corresponding nanopore experiments would be valuable for the article.

We have included a supplementary figure (Supplementary Figure 25) and a supplementary table (Supplementary Table 6) result of the suggested experiment. We have performed RNA ID assembly of the transcription products of a 4.5 kbp DNA construct where the OriC sequence is further separated from the T7RNAP promoter. Nanopore experiments agree with agarose gels presented in Supplementary Figure 22.

Figure 25

Figure 25: Single molecule sizing of RNA IDs produced from a 4.5 kbp DNA construct. **a** The sequence of the DNA construct is presented in Supplementary Table 4. The linearized

version of the construct (Drall) is presented Supplementary Figure 22, in lane 11, and the transcription products are shown in lane 12. The RNA ID design includes an 'R' label and two '1' bits. The oligos used for assembly of the hybrid are shown in supplementary Table 6. **b** Exemplary RNA ID translocation events of full-length transcripts (END). **c** Exemplary RNA ID translocation events ascribed to premature termination (PT). **d** Scatter plot of mean current against translocation time shows two distinct distributions, attributed to PT RNA IDs and END RNA IDs (from left to right). **e** Scatter plot of mean current against translocation time of 100 unfolded events. **f** Scatter plot of charge deficit against translocation time for RNA IDs of PT (red) and END (gray) RNA transcripts, which shows the linear dependence of both parameters. **g** Base pair length of molecules converted from translocation time (in f), which shows two distinct distributions. PT distribution has a mean length of (2.9 ± 0.2) kbp and END transcripts have a mean length of (4.3 ± 0.3) kbp. Errors correspond to standard deviation.

Table of oligos used:

Table 6.

Sequence of oligonucleotides used to assemble RNA ID for characterization of transcripts originating from 4.5 kbp DNA construct. This construct has a larger separation between T7 promoter and OriC (Supplementary Table 4). This design is used for the experiments presented in Supplementary Figure 25.

Table 6

Oligo number	Sequence (5' → 3')
1	AGCGTGATGCTACTAATTGGGACAATTTTCCAGATGAAGT
2	ATCATCTAAGAATTTAAATGAAGAAGACTTCAGAGCTTTT
3	GTAAAAAATTATTTGGCAAAAATAATATAATTCCGGCTGCA
4	GGGGCGCCTCGTGATACGCCTATTTTTATAGGTTAATGT
5	CATGATAATAATGGTTTCTTAGACGTCAGGTGGCACTTTT
6	CGGGGAAATGTGCGCGGAACCCCTATTTGTTTATTTTTCT
7	AAATACATTCAAATATGTATCCGCTCATGAGACAATAACC
8	CTGATAAATGCTTCAATAATATTGAAAAAGGAAGAGTATG
9	AGTATTCAACATTTCCGTGTCGCCCTTATCCCTTTTTTG
10	CGGCATTTTGCCTTCCTGTTTTTAAAGAAGTTTGACCCAG
11	AAACGCTGGGAAAGGCGGTAAAATATGCACGAAGATCAGT
12	TCGGTTCACGAGTGGGTACATCGAACTGGATCTCAACAG
13	CGGTAAGATCCTTGAAGAGTTTTCCGCCCGAAGAACGTT
14	TTCCAATGATGAGCACTTTTAAAGCTCTGCTACTGTGGCG
15	CGGTTATATCCCGTATTGACGCCGGGCAAGAGCAACTCGG
16	TCGCCGCATACACTATTCTCAGAATGACTTGGTTGAGTAC
17	TCACCAGTCACAGAAAAGCATCTTACGGATGGCATGACAG
18	TAGAGAATTATGCAGTGCTGCCATAACCATGAGTGATAAC
19	ACTGCGGCCAACTTACTTCTGACAACGATCGGAGGACCGA
20	AGGAGCTAACCGCTTTTTTGCACAACATGGGGGATCATGT

21	AACTCGCCTTGATCGTTGGGAACCGGAGCTGAATGAAGCC
22	ATACCAAACGACGAGCGTGACACCACGATGCCTGTAGCAA
23	TGGCAACAACGTTGCGCAAACCTATTAACCTGGCGAACTACT
24	TACTCTAGCTTCCCGGCAACAATTAATAGACTGGATGGAG
25	GCGGATAAAGTTGCAGGACCACTTCTGCGCTCGGCCCTTC
26	CGGCTGGCTGGTTTATTGCTGATAAATCTGGAGCCGGTGA
27	GCGTGGGTCTCGCGGTATCATTGCAGCACTGGGGCCAGAT
28	GGTAAGCCCTCCCGTATCGTAGTTATCTACACGACGGGGA
29	GTCAGGCAACTATGGATGAACGAAATAGACAGATCGCTGA
30	GATAGGTGCCTCACTGATTAAGCATTGGTAACTGTCAGAC
31	CAAGTTTACTCATATATACTTTAGATTGATTTAAAACTTC
32	ATTTTTAATTTAAAAGGATCTAGGTGAAGATCCTTTTTGA
33	TAATCTCATGACCAAAAATCCCTTAACGTGAGTTTTTCGTTC
34	CACTGAGCGTCAGACCCCGTAGAAAAGATCAAAGGATCTT
35	CTTGAGATCCTTTTTTTCTGCGCGTAATCTGCTGCTTGCA
36	AACAAAAAAACCACCGCTACCAGCGGTGGTTTGTTCGCCG
37	GATCAAGAGCTACCAACTCTTTTTCCGAAGGTAACGGCT
38	TCAGCAGAGCGCAGATACCAAATACTGTTCTTCTAGTGTA
39	GCCGTAGTTAGGCCACCACTTCAAGAACTCTGTAGCACCG
40	CCTACATACTCGCTCTGCTAATCCTGTTACCAGTGGCTG
41	CTGCCAGTGGCGATAAGTCGTGTCTTACCGGGTTGGACTC
42	AAGACGATAGTTACCGGATAAGGCGCAGCGGTCTGGGCTGA
43	ACGGGGGGTTCGTGCACACAGCCCAGCTTGGAGCGAACGA
44	CCTACACCGAACTGAGATACCTACAGCGTGAGCTATGAGA
45	AAGCGCCACGCTTCCCGAAGGGAGAAAGGCGGACAGGTAT
46	CCGGTAAGCGGCAGGGTCGGAACAGGAGAGCGCACGAGGG
47	AGCTTCCAGGGGAAACGCCTGGTATCTTTATAGTCCTGT
48	CGGGTTTCGCCACCTCTGACTTGAGCGTCGATTTTTGTGA
49	TGCTCGTCAGGGGGGCGGAGCCTATGAAAAACGCCAGCA
50	ACGCGGCCTTTTTACGGTTCCTGGCCTT
51	TTGCTGGCCTTTTGTCTCACATGTTCTTTC
52	CTGCGTTATCCCCTGATTCTGTGGATTGGATATCACTCATTAGTGGT
53	TAACCGTATTACCGCCTTTGAGTGAGCTGATACCGCTCGC
54	CGCAGCCGAACGACCGAGCGCAGCGAGTCAGTGAGCGAGG
55	AAGCGGAAGAGCGCCAATACGCAAACCGCCTCTCCCCGC
56	GCGTTGGCCGATTTCATTAATGCAGCTGGCACGACAGGTTT
57	CCCGACTGGAAGCAATTGGCAGTGAGCGCAACGCAATTA
58	ATGTGAGTTAGCTCACTCATTAGGCACCCCAGGCTTTACA
59	CTTTATGCTTCCGGCTCGTATAATGTGCTGATGAATCCCC
60	TAATGATTTTGGTAAAAATCATTAAGTTAAGGTGGATACA
61	CATCTTGTCATATGATCAAATGGTTTCGCGAAAAATCAAT
62	AATCAGACAACAAGATGTGCGAACTCGATATTTTACACGA
63	CTCTCTTTACCAATTCTGCCCGAATTACACTTAAAACGA
64	CTCAACAGCTTAACGTTGGCTTGCCACGCATTACTTGACT
65	GTAAACTCTCACTCTTACCGAACTTGCCCGTAACCTGCC
66	AACCAAAGCGAGAACAAAACATAACATCAAACGAATCGAC

67	CGATTGTTAGGTAATCGTCACCTCCACAAAGAGCGACTCG
68	CTGTATACCGTTGGCATGCTAGCTTTATCTGTTCCGGCAA
69	TACGATGCCCATTTGTACTTGTGACTGGTCTGATATTCGT
70	GAGCAAAAACGACTTATGGTATTGCGAGCTTCAGTCGCAC
71	TACACGGTCGTTCTGTTACTCTTTATGAGAAAGCGTTCCC
72	GCTTTCAGAGCAATGTTCAAAGAAAGCTCATGACCAATTT
73	CTAGCCGACCTTGCGAGCATTCTACCGAGTAACACCACAC
74	CGCTCATTGTCAGTGATGCTGGCTTTAAAGTGCCA
75	TGGTATAAATCCGTTGAGAAGCTGGTTTGGATATCACTCATTAGTGGT
76	GTTGGTACTGGTTAAGTCGAGTAAGAGGAAAAGTACAATA
77	TGCAGACCTAGGAGCGGAAAACTGGAAACCTATCAGCAA
78	CTTACATGATATGTCATCTAGTCACTCAAAGACTTTAGGC
79	TATAAGAGGCTGACTAAAAGCAATCCAATCTCATGCCAAA
80	TTCTATTGTATAAATCTCGCTCTAAAGGCCGAAAAAATCA
81	GCGCTCGACACGGACTCATTGTCACCACCCGTCACCTAAA
82	ATCTACTCAGCGTCGGCAAAGGAGCCATGGGTTCTAGCAA
83	CTAACTTACCTGTTGAAATTCGAACACCCAAACAACCTTGT
84	TAATATCTATTTCGAAGCGAATGCAGATTGAAGAAACCTTC
85	CGAGACTTGAAAAGTCTGCCTACGGACTAGGCCTACGCC
86	ATAGCCGAACGAGCAGCTCAGAGCGTTTTGATATCATGCT
87	GCTAATCGCCCTGATGCTTCAACTAACATGTTGGCTTGCG
88	GGCGTTCATGCTCAGAAACAAGGTTGGGACAAGCACTTCC
89	AGGCTAACACAGTCAGAAATCGAAACGTACTCTCAACAGT
90	TCGCTTAGGCATGGAAGTTTTGCGGCATTCTGGCTACACA
91	ATAACAAGGGAAGACTTACTCGTGGCTGCAACCCTACTAG
92	CTCAAAATTTATTCACACATGGTTACGCTTTGGGGAAATT
93	ATGAGGGGATCTCTCAGTATAATGTGTGGAATTATGAGCG
94	GATAATAATTTACACAGGAGGTTTAAACTTTAAACATGT
95	CAAAAGAGACGTCTTTTGTAAAGAATGCTGAGGAACTTGC
96	AAAGCAAAAAATGGATGCTATTAACCCTGAACTTTCTTCA
97	AAATTTAAATTTTTAATAAAATTCCTGTCTCAGTT
98	TCCTGAAGCTTGCTCTAAACCTCGTTTTGGATATCACTCATTAGTGGT
99	TCAAAAAAATGCAGAATAAAGTTGTTTGGATATCACTCATTAGTGGT
100	GTCAAGAGGAACATATTGAATATTTAGCTCGTAGTTTTCA
101	TGAGAGTCGATTGCCAAGAAAACCCACGCCACCTACAACG
102	GTTCCCTGATGAGGTGGTTAGCATAGTTCTTAATATAAGTT
103	TTAATATACAGCCTGAAAATCTTGAGAGAATAAAAGAAGA
104	ACATCGATTTTCCATGGCAGCTGAGAAATATTGTAGGAGAT
105	CTTCTAGAAAGATTTAAGCCG
106	AGAATGGTCTGTGATCCCCC
107	CATTCCCGGCTACACTGCACCATGATCTTGCTGAAAAACT
108	CGAGCCATCCGGAAGATCTGGCGGCCGCTCTCCC
109	CAGCAGCAGCAGCAGCAGCAGCAGCAGCAGCAG

The proportion of “good” events (unfolded and showing the ‘R’ marker and the ‘1’ bits) compared to the total number of events recorded should be provided for a sake of transparency about the difficulty of the experiment. This would avoid vague expression about the representativity of the display event (line 368).

To address this comment, we have included a new supplementary figure (Supplementary Figure 27) which includes the percentage of folded events, unfolded events and the events used for sizing. We also present some folded events so that the reader can see a full range of data.

Figure 27

Figure 27: Percentage of translocation events sized. **a** Shows the percentage of folded, unfolded and sized events. The percentages were computed from 3 different nanopore measurements of RNA IDs produced from transcription of a circular DNA construct. The amount of folded events correspond to $(73 \pm 2)\%$, unfolded events constitute $(27 \pm 2)\%$ of the sample, and $(24 \pm 3)\%$ were sized. The errors correspond to the standard error of the mean. **b** Exemplary folded events are presented.

Moreover, we have included a supplementary figure (Supplementary Figure 28) that describes in detail the data analysis performed.

Figure 28

Figure 28: Step-by-step nanopore data analysis for RNA ID sizing. **a** An exemplary region of the ionic raw current trace is shown. We use a home-built LabVIEW code to identify translocation events by simple thresholding. Events are then analysed by using standard parameters, which include the charge deficit, minimum translocation time and mean current that are given by the length of the RNA ID molecule and its design. **b** The translocation events were then categorized into PT RNA ID, END RNA ID, folded RNA ID

and translocations associated to misfolded molecules (marked X) based on the ionic current trace of the events. **c** The base pair length of PT RNA IDs and END RNA IDs was computed using the RNA ID design. The distance between current spikes (blue) was measured (in time units). This distance corresponds to a known base pair length which was used to calculate a conversion factor. Then, the length in time units of the entire translocation event was measured (pink) and converted into a base. The base pair length was plotted as shown in the right.

Minor comments

On line 146, "which revealed" should be replaced by "which confirmed" as the two types of RNA were revealed earlier in the text using gel electrophoresis.

We have replaced the suggested phrase (page 4, line 151):

'Through the inclusion of labelled oligos, RNA IDs enabled the characterization of RNA transcripts by nanopore sensing, **which confirmed** two types of RNA after transcription.'

On line 165, in the sentence "Besides, the rest of the RNA is decorated with '1' bits" should be rephrased. It is indeed misleading (only the beginning of the RNA is decorated. The END event on figure 1e shows exactly that there is no decoration on the second half of the RNA.

We have rephrased the suggested sentence (page 5, line 171):

'Besides, **parts of the RNA are** decorated with '1' bits (Figure 1c).'

The translation of translocation time to RNA length assumes that the speed is constant along one translocation. In the $n > 2$ experiments, the time between 'R' markers should provide some insight about this.

The variation in translocation speed along a single DNA translocation has been widely studied previously (Chen et al, Nature Physics, 2021). It has been observed that there exists a small variability (<10%) in translocation speed throughout nucleic acids of ~7kbp. Followed by an increase in translocation speed at the end of the construct. We have included a statement to address how this influences our experiments (page 9, line 259).

Variability in transcript sizing due to changes in translocation speed along an RNA ID molecule⁴¹ are within error.'

On line 345 about figure 3e, the plot does not show a linear dependence with 'n' but rather a linear dependence of the charge deficit with the translocation time. These informations are provided in figure 17b (and are in my opinion, more interesting than the figure 3e linear dependency).

We thank the author for this comment. We consider that Figure 3e does illustrate the increase in event charge deficit and translocation time as the number of transcription

cycles increases. We consider that Figure 3e contains the information provided in Figure 17b while providing a description of the translocation time of the transcripts. We have however rephrased our statement in page 14, line 357, to emphasize your point:

'As can be observed from **Figure 3e**, translocation events of larger RNA IDs, produced from more transcription cycles, have larger translocation times, because longer molecules take more time to translocate through the pore. For the same reason, the charge deficit increases in events associated to larger transcript sizes (**Figure 3e**). Both the translocation time and charge deficit are seen to be linearly correlated.'

The pore size should be provided for all data presented.

We have now included a supplementary Table 8 containing the IV curves and RMS noise for the nanopore experiments performed and derived an estimate of each nanopore diameter.

Table 8.

The table shows the IV curves of nanopores used to study RNA IDs. The ionic current and RMS noise at 600 mV is presented for each pore. An estimate of each nanopore diameter was calculated from the ionic current values presented, assuming a conical pore geometry⁵.

Table 8

Pore number	Sample	Current, RMS noise at 600 mV and calculated pore diameter	Current / voltage curve (IV curve)
1	RNA IDs produced from linear DNA	11.4 nA 6.5 pA ~8 nm	
2 RNA IDs produced from linear DNA, and DNA template

9.9 nA
6.7 pA
~7 nm

3 RNA IDs of produced from circular template

8.7 nA
6.2 pA
~6 nm

4 RNA IDs of produced from circular template

7.5 nA
6.6 pA
~5 nm

5

RNA IDs
of produced
from circular
template

12.2 nA
6.4 pA
~9 nm

6

RNA IDs
of produced
from circular
template

10.0 nA
6.4 pA
~7 nm

7

DNA ladder
SF8
(0.5 kbp –
10 kbp)

11.1 nA
5.7 pA
~8 nm

8

RNA IDs
produced
from 4.5
kbp DNA

10.9 nA
6.4 pA
~8 nm

An estimate the pore's diameter was calculated from the overall resistance of the nanopore in open state, R , which is constituted by the resistance of the pore cavity region, R_{pore} , and the resistance of the access region of the pore, R_{acc} ⁵:

$$R = R_{pore} + R_{acc}$$

This equation can be rewritten in terms of the resistivity ρ of the electrolytic solution, the pore's length, L , and the diameter of the *cis* and *trans* aperture of the pore, D_{cis} and D_{trans} :

$$R = \rho \frac{4L}{\pi D_{trans} D_{cis}} + \rho \left(\frac{1}{2 D_{trans}} + \frac{1}{2 D_{cis}} \right)$$

The diameter of the pore D_{cis} was calculated using the experimental ionic current I during the application of a 600 mV potential, assuming a D_{trans} of 200 μm , conductivity of 15.5 Sm^{-1} for 4M LiCl, and length L of 950 μm for our glass nanopores^{6,7}.

In supplementary figure 5, there is an inconsistency of the total number of events in the figure and its legend. Figure 5a says 1500 events and the legend says 1700 events.

Thank you for pointing this out, we have corrected it, the text in the figure caption is now in agreement with the legend and the Figure:

Figure 5

Figure 5. Characterization of RNA ID using charge deficit, mean current and translocation time. **a** Histogram of the charge deficit of RNA ID still in the presence of the linear DNA template. Histogram shows 3 distributions, the one with the lowest charge deficit is ascribed to premature termination (PT), the middle distribution corresponds to transcription of the full linear DNA (END) and the distribution furthest to the right is ascribed to the linear DNA template. This distribution (composed of 1500 events) includes translocations of molecules with multiple conformations, which include folded events, constructs with knots and unfolded events. **b** After treatment with DNase I, it can be seen how the distribution furthest to the right, ascribed to DNA, is removed. Unfolded translocations of both PT (red) and END (gray) RNA IDs (presented in Figure 2c) describe the entire sample, despite their conformation, while enabling single-molecule sizing. These distributions correspond to the same nanopore measurement presented in Figure 2. **c** Scatter plot of mean current against translocation time shows three distinct distributions, attributed to PT RNA IDs, END RNA IDs, and the linear template (from left to right). Unfolded molecules take longer to translocate through the pore than folded molecules but cover less cross-sectional area of the pore while translocating, producing a less significant drop in ionic current for longer times. Plotting events with different conformations produces this type of distribution, with unfolded events at the top, and folded events at the bottom of each distribution. **d** Selection of unfolded events (in the

sample treated with DNase I) is performed to describe each distribution. Unfolded events were found at the top of each distribution, and they are representative of the whole sample.

Add the proportion of 'unfolded event'

We have included this in the supplementary Figure 27.

Figure 27

Figure 27: Percentage of translocation events sized. **a** Shows the percentage of folded, unfolded and sized events. The percentages were computed from 3 different nanopore measurements of RNA IDs produced from transcription of a circular DNA construct. The amount of folded events correspond to $(73 \pm 2)\%$, unfolded events constitute $(27 \pm 2)\%$ of the sample, and $(24 \pm 3)\%$ were sized. The errors correspond to the standard error of the mean. **b** Exemplary folded events are presented.

In Supplementary figure 5b the red and grey distribution seems to be the one presented on figure 2c. Is this correct? please clarify.

This is correct, we have now clarified it in the figure caption and labelled more clearly the populations in the Figures.

Figure 5

Figure 5. Characterization of RNA ID using charge deficit, mean current and translocation time. **a** Histogram of the charge deficit of RNA ID still in the presence of the linear DNA template. Histogram shows 3 distributions, the one with the lowest charge deficit is ascribed to premature termination (PT), the middle distribution corresponds to transcription of the full linear DNA (END) and the distribution furthest to the right is ascribed to the linear DNA template. This distribution (composed of 1500 events) includes translocations of molecules with multiple conformations, which include folded events, constructs with knots and unfolded events. **b** After treatment with DNase I, it can be seen how the distribution furthest to the right, ascribed to DNA, is removed. Unfolded translocations of both PT (red) and END (gray) RNA IDs (presented in Figure 2c) describe the entire sample, despite their conformation, while enabling single-molecule sizing. These distributions correspond to the same nanopore measurement presented in Figure 2. **c** Scatter plot of mean current against translocation time shows three distinct distributions, attributed to PT RNA IDs, END RNA IDs, and the linear template (from left to right). Unfolded molecules take longer to translocate through the pore than folded molecules but cover less cross-sectional area of the pore while translocating, producing a less significant drop in ionic current for longer times. Plotting events with different

conformations produces this type of distribution, with unfolded events at the top, and folded events at the bottom of each distribution. **d** Selection of unfolded events (in the sample treated with DNase I) is performed to describe each distribution. Unfolded events were found at the top of each distribution, and they are representative of the whole sample.

We could expect to see an equal number of 3'->5' and 5'->3' events. But the 'representative' (line 368) events in the supplementary figure. For instance the SFigure 12 shows only 5'->3' events... is this really representative? Do the author have an explanation of this if yes.

It is indeed expected that the 3'->5' and 5'->3' events detected is similar. We included most events in one direction to make it easier for the reader to visualize the current spikes and associate them to their respective label. We have now included events translocating in the opposite direction in Supplementary Figure 16c to provide a better representation of the sample.

Figure 16.

Figure 16. $N = 5$ and $N > 5$ example events. **a** Example of nanopore events of RNA IDs produced from five transcription cycles ($N = 5$) measured in pores with different sizes. All measurements were performed under the same applied voltage of 600 mV. **b** Also, RNA IDs with $N > 5$ were identified, example events are shown from different nanopore measurements. **c** Finally, this figure shows translocation of RNA IDs for each transcription cycle that entered to the pore in the 5' to 3' direction.

We have also indicated this to the reader in the main text (page 14, line 343):

“RNA IDs can translocate both 5'-3' and 3'-5' directions through the nanopore. These events show translocations in the 3'-5' direction. Events translocating in the opposite direction are shown in Supplementary **Figure 16.**”

SFigure 18, is there a specific meaning to invert the order of the plots in 18a part compare to 18b part?

No, it was only a design decision, we have modified it for unfolded translocations to be on top in figure 18a to avoid confusion for the readers.

Figure 18. Unfolded (linear) RNA IDs events are representative of the events with different conformations. **a** Histogram of charge deficit for all translocations detected in one nanopore measurement (black, 2000 events). The selection of 219 unfolded events shows a distribution of charge deficit which is representative of the distribution of all translocations detected, therefore these events can be used to describe the sample and gain single-molecule information from the RNA ID design. **b** Scatter plot of mean current against translocation time, which also shows that the selection of unfolded can be used for the description of a sample within a defined parameter space.

The verb 'to enable' is used about 30 times throughout the main text and especially 4 times between line 75 and line 90. The author should consider synonyms to avoid reading's heaviness.

We've modified our manuscript to avoid repetitiveness and to improve readability.

Reviewer #3 (Remarks to the Author):

What are the noteworthy results?

The paper showcases the application of the technology of RNA-ID developed at the Keyser lab to studying in vitro RNA transcription with T7 Polymerase. In the paper, authors encounter premature transcription termination that happens at OriC with around 50% probability. The authors use RNA ID technology in conjunction with nanopore readout, offering a more cost-effective alternative to traditional methods like RNE-seq. This approach also overcomes the challenges associated with performing long RNA reads.

Will the work be of significance to the field and related fields? How does it compare to the established literature? If the work is not original, please provide relevant references.

The work is original, and the fact of premature transcription termination is both unexplored and very intriguing. The work definitely represents an asset for in vitro studies of DNA transcription by various proteins and their modifications. The significance of the work spreads into the field of microbiology, biophysics, single-molecule biophysics, and also shows great utility for the nanopore sensing platform.

Does the work support the conclusions and claims, or is additional evidence needed?

The claim that nanopore technology combined with RNA ID tagging can be used to study transcription is fully supported by the work. The claim of studying the T7 polymerase transcription prompts some questions, that could be addressed in this cycle of revision:

We appreciate the reviewer's detail assessment of our manuscript and for acknowledging that our study 'overcomes the challenges associated with performing long RNA reads'. We thank the reviewer for describing the work as original and acknowledging that studying RNA ID with nanopore technology and RNA ID tagging is fully supported by our data. Below, we replied to the reviewer's points with the corrections that we implemented.

1)Is only the OriC terminating the transcription prematurely, or do other Origins trigger that? (I think if experiments are needed to answer this question, then they don't belong not the scope of this paper, but maybe indicated in tes)

We have included a brief discussion to address the reviewer's comment (page 18, line 452):

'Transcription termination of bacterial RNA polymerases in OriC has been identified in multiple bacterial systems⁴⁶. For this reason, exploring the effect of origins of replication in transcription could offer valuable insights into the understanding of transcript diversity.'

I am missing a discussion. Why, if OriC terminates the transcription, does the entire transcript of OriC end up in the RNA? 2)Is there a part of the OriC that acts as a terminator, or is it the entire sequence? (this question I think can be well addressed with the methodology developed, or at lease the prospect of it should be indicated in text. I don't think extra experiments are needed to publish this paper).

We've included the suggested discussion in the main text (page 17, line 441):

'We conclude premature transcription termination from the location of the second '1' bit in the RNA ID design, which is positioned 0.2 kbp downstream of the beginning of the OriC sequence. The '1' bit produces a current spike at the end of the translocation event, indicating that rho independent termination occurs downstream of the '1' bit. A sequence (Supplementary Table 7) downstream and in close proximity to the '1' bit shows structural similarity to T7RNAP terminators previously reported⁴⁵, which makes it a potential candidate responsible for the reported termination.'

Supplementary Table 7 mentioned in the included discussion:

Table 7.

Identified sequence within DNA construct that shows structural similarity to engineered T7RNAP transcription terminators.

Table 7

DNA	Sequence (5' → 3')
1	CAAACAAACCACCGCTGGTAGCGGTGGTTTTTTTTGTTT

3)Figure 3d contains a gel where in the first lane we have RNA product of circular rolling transcription and the lines in it indicate that it's most often terminated in a certain spot all the time. But I see faint lines at around 3kb (between n=2 and n=1) and maybe (not sure because of the color boxes) between n=2 and n=3. It would be good to comment on those, and if those are true RNA products, I would expect that there should be events, which don't only end with R11 ID, but extend further.

These faint bands can indeed be observed in our gels. You can also see them in Supplementary Figure 10 and Supplementary Figure 22 (lanes 3 and 4). Nanopore measurements reveal that these bands correspond to real RNA transcripts; for instance, in Figure 3d or Figure 4d, the event charge deficit, translocation time and base pair length distributions show population that can be attributed to these bands. We attribute these low abundance bands (and distributions) to fall of RNA polymerase.

We have now discussed this in the main text, in Page 16 (line 414), section where we elucidate the meaning of each of the distributions:

'The less prominent distributions, located in between the main populations associated to premature termination, at ~3.1 kbp for n = 1 and ~5.9 kbp for n = 2, are ascribed to dissociation of the T7RNAP. These agree with the faint bands in agarose gels of the same transcript sizes presented in **Figure 3d**.'

4)This question stems more or less from the previous one: Were there any prematurely terminated events not at the origin? There should be some probability of it happening.

As shown before, there are events that are not prematurely terminated at the origin. These represent a small population of the sample and are described in detail in Supplementary Figure 26. We have enriched the discussion (line 126) to indicate this to the readers:

'T7RNAP was engineered to be a processive enzyme, hence we expect low dissociation without a defined terminator sequence⁸.'

5)The authors derive the probability of dissociation from the OriC based on first experiments with linear DNA, but I think based on the rolling transcription it is possible to calculate the entire binomial statistics and also deduce this probability. It would be good to compare both.

Yes, it is possible. We have included a supplementary figure (Supplementary Figure 26) quantitative description of T7RNAP processivity and transcription termination. In the analysis performed we account for the probability of termination at the terminator and dissociation in a different region of the plasmid. The derived termination probability at the terminator agrees with the termination probability obtained for linear DNA, and as expected, the fall-off probability in a different position is low.

Figure 26

Figure 26: Quantitative description of T7RNAP processivity and transcription termination. **a** Considering a probability p of transcription terminating at the identified premature transcription termination site, and a probability p_s for transcription terminating solely by dissociation of the polymerase at a different region of the plasmid. Equation $f(x)$ describes the abundance of transcripts originated from premature transcription termination at OriC. Equation $g(x)$ describes the transcript abundance originated from fall-off of T7RNAP at a different region of the plasmid. **b** The distribution of the charge deficit of nanopore

translocation events is presented. The different transcript populations were fitted to gaussian functions, from which the relative abundance of transcripts was derived. The distributions ascribed to premature termination transcripts are labelled with x values of odd integers ($x = 1, 3, 5, 7$) and the minor distributions ascribed to transcripts produced from T7RNAP dissociation in a different region of the plasmid receive x values of even integers ($x = 2, 4, 6, 8$). **c** The relative abundance of transcripts (black) are plotted. Fitting of $f(x)$ and $g(x)$ is plotted in red, $p \sim 0.51$ and $p_s \sim 0.21$, which agrees with transcription termination reported in linear DNA constructs.

Reviewer 3 (Additional Document):

We appreciate the time the reviewer took to point out specific comments within the manuscript, we addressed those comments in the following way:

To address the multiple comments made by the reviewer on the paragraph beginning in line 52 of our manuscript, we have increased the rigour of the paragraph (line 52, page 1).

'For the development of novel RNA technologies and proficient characterization of gene expression, in-depth understanding of transcript diversity is required. Accurate assessment of transcript size and heterogeneity is needed, as well as precise identification of splicing patterns and transcript variants. The main approaches to studying RNA are based on bulk techniques such as gel electrophoresis, quantitative reverse transcription–polymerase chain reaction (qRT-PCR) and RNA sequencing (RNA-seq)¹¹. Nevertheless, these techniques may face limitations in understanding RNA diversity in its native form. qRT-PCR requires additional enzymatic steps and may encounter reverse transcription and polymerase biases¹². The same happens with RNA-seq which may also face short-read limitations¹³. Despite its practical simplicity, accurate analysis of agarose gels can be complicated as no information about the sequence is obtained and electrophoretic mobility can be affected by the conformation of RNA¹⁴. Unexpected transcription species in gels may indicate transcription initiation in an alternative promoter sequence or they may correspond to the same RNA product with another conformation and therefore different electrophoretic mobility. Considering the capabilities of current RNA characterization methods, a robust platform is needed, which can provide accurate description of transcript heterogeneity and retains the native diversity of RNA transcripts.

Single-molecule approaches have been employed to study both RNA transcripts and RNAPs themselves.'

Looks like a comma would be better

We substituted the suggested ',' (page 2, line 73).

By combining optical tweezers and fluorescent microscopy, the dynamics of RNAP at each step of transcription has been investigated¹⁷, showing that single-molecule techniques can provide further insight into the characterization of transcription¹⁸.

In this part of the introduction before I have read the paper, it is totally unclear how your strategy "illustrates transcripton".

We have rephrased paragraph starting in page 2, line 89 so that the reader can easily understand that we identify a premature transcription termination site:

'Our strategy enables the quantitative description of the T7RNAP's ability to continuously transcribe different DNA templates **and identifies premature transcription termination sites using sequence-specific labelling.**'

It is not clear what strategies were used to prevent the nanopore readout variability.

Performing nanopore measurement of PT RNA IDs and END RNA IDs in the same pore prevents readout variability. For example if PT RNA IDs and END RNA IDs were measured in different pores event parameters would not be fully comparable.

Feels ambiguous because even without labels you decorate it with complementary DNA oligos

We have rephrased, as suggested the statement in line 195, page 6:

In END RNA IDs, the nanopore translocation shows the 'R' and '1' bit current spikes, followed by a prolonged plateau which is attributed to the region with no streptavidin labels in the RNA ID, only constituted by RNA-DNA duplex. The plateau correlates to the region spanning from the OriC sequence to the end of the linear DNA template.

Would be good to see where exactly on OriC it withdraws it

We now discuss the reviewer's comment on the position where the polymerase terminates in the discussion section (page 17, line 441):

We conclude premature transcription termination from the location of the second '1' bit in the RNA ID design, which is positioned 0.2 kbp downstream of the beginning of the OriC sequence. The '1' bit produces a current spike at the end of the translocation event, indicating that rho independent termination occurs downstream of the '1' bit. A sequence (Supplementary Table 7) downstream and in close proximity to the '1' bit shows structural similarity, which makes it a potential candidate responsible for the reported termination.

They are really redundant, 2a is not needed, also, given that there is also explanation in 2b and 2c.

Regarding redundancy of Figure 1e and 2a commented by the reviewer in line 218. We understand that the reviewer may find the figures redundant, nevertheless, we would like to keep Figure 2a. Previous feedback we have received from the manuscript suggests that it is helpful for identifying PT and END distributions in the provided histograms.

In response to reviewer's request on showing folded events, we have included folded events in supplementary figure 27 so that the reader can visualize some of them.

Figure 27: Percentage of translocation events sized. **a** Shows the percentage of folded, unfolded and sized events. The percentages were computed from 3 different nanopore measurements of RNA IDs produced from transcription of a circular DNA construct. The amount of folded events correspond to $(73 \pm 2)\%$, unfolded events constitute $(27 \pm 2)\%$ of the sample, and $(24 \pm 3)\%$ were sized. The errors correspond to the standard error of the mean. **b** Exemplary folded events are presented.

The distribution in kb can be narrower because you should take into account the speeding up of the translocation towards the end.

Regarding the reviewers comment on translocation velocity made in line 239. The variation in translocation speed along a single DNA translocation has been widely studied previously (Chen et al, Nature Physics, 2021). It has been observed that there exists a small variability ($<10\%$) in translocation speed during a translocation event of $\sim 7\text{kbp}$. Followed by an increase in translocation speed at the end of the construct. We have included a statement to address how this influences our experiments (page 9, line 259). This intra-event variability should not be confused with the overall variance in translocation time between events.

Variability in transcript sizing due to changes in translocation speed along an RNA ID molecule⁴¹ are within error.'

There is a lot happening in the lanes for the plasmid, which is worth explaining.

DNase I treatment and the lane of plasmids are now explained in more detail in Supplementary Figure 10. The multiple bands are ascribed to the different physical configurations of supercoiled DNA while being electrophoretically driven through the agarose gel (Gibson et al, DNA Electrophoresis: Methods and Protocols, 2020). Nanopore sensing (and restriction cutting with *Dralll*) confirmed a unique DNA molecule. Figure 9 shows the charge deficit of a supercoiled DNA plasmid translocated through a nanopore. The plasmid is shown in Supplementary Figure 22, lane 1. A unimodal distribution of the charge deficit is observed.

Figure 10.

Figure 10. Rolling circle transcription of circular DNA construct. DNA ladder and ssRNA ladder are included on both sides of the gel. Lane 1 – Circular plasmid with 12 CTG repeats (sequence of circular plasmid in Supplementary Table 1). The multiple bands are ascribed to the physical configurations that supercoiled DNA has while being electrophoretically driven through the agarose gel. Lane 2 – circular plasmid from lane 1 treated with *Escherichia coli* Topoisomerase I to induce plasmid relaxation. Lane 3 – RNA from transcription of Topoisomerase I treated circular plasmid in lane 2. Lane 4 – RNA from lane 3 treated with DNase I. From DNase I treatment, DNA band located at ~3 kbp (DNA) is removed. The rest of the bands, ascribed to RNA products of the multiple transcription cycles, remain.

Figure 9.

Figure 9. The 3.1 kbp circular DNA used as the template for rolling circle transcription was characterized using nanopore sensing. The charge deficit of the translocation events shows a unimodal distribution, demonstrating the presence of a single DNA construct. This also confirms RNA is produced from circle rolling transcription of the 3.1 kbp circular DNA, and not by transcription of DNA dimers or trimers of the 3.1 kbp DNA which could be a possible interpretation of agarose gel electrophoresis. Here we also showcase the clarity nanopore sensing provides for the elucidation of DNA identity.

It is unclear what is the high processivity of T7RNAP compared to, even considering the reference.

We have rephrased our claim in page 11, line 322 to increase its precision, as suggested by comment reviewer:

With our strategy, we observed RNA IDs originating from one (blue), two (orange), three (green), four (red), or five (purple) transcription cycles (Figure 3c), **showcasing capability of T7RNAP to synthesize transcripts in the kilobase range⁴³**.

To respond to the reviewer's comment made on figure 3d we included the following statement in page 16, line 414, where we discuss the meaning of each distribution observed both in nanopores and agarose gels:

'The less prominent distributions, located in between the main populations associated to premature termination, at ~3.1 kbp for $n = 1$ and ~5.9 kbp for $n = 2$, are ascribed to dissociation of the T7RNAP. These agree with the faint bands in agarose gels of the same transcript sizes presented in **Figure 3d.'**

Finally, in figures and text, the abbreviation of kilobase pairs was changed to (kbp) to distinguish from kilobases (kb) in single-stranded molecules. Furthermore, the order of the supplementary tables has been rearranged so that they are all mentioned chronologically in the main text of the manuscript.

REVIEWERS' COMMENTS

Reviewer #1 (Remarks to the Author):

In this article, the authors have conducted a quantitative analysis of alternative transcription termination and the transcription capabilities of RNA polymerase through the innovative use of RNA ID and quartz capillary nanopores. This method facilitates the direct differentiation of various full-length RNA transcripts at the single-molecule level, marking a significant advancement in related research fields. It offers novel perspectives for investigating transcription termination mechanisms.

Following the suggestions of peer reviewers, the authors have conducted additional experiments to substantiate their hypotheses. Comprehensive experimental data supporting their findings are thoroughly presented in the supplementary information. As a result, the work robustly underpins the drawn conclusions and assertions. The authors have meticulously incorporated specific standards and criteria for selecting translocation signals derived from the transcription processes. Moreover, the analytical procedures have been clearly articulated. The inclusion of the formula used to calculate the pore size of quartz nanopores is a commendable addition, lending credence to the data analysis, interpretation, and ensuing conclusions.

The assertion that nanopore technology, when integrated with RNA ID tagging, is effective for studying transcription processes is convincingly supported by this study. The work demonstrates that the combination of nanopore detection and RNA ID design is a viable approach for examining alternative transcription termination.

The methods section is detailed and comprehensive. I concur that the manuscript merits publication due to its methodological rigor and the significance of its contributions to the field.

Reviewer #2 (Remarks to the Author):

The author have convincingly answer all the remarks pointed out and I thank them for their objectivity and the precision of the answers.

With all these corrections, the article is should accepted for publication.

Jérôme Mathé

Reviewer #3 (Remarks to the Author):

I am satisfied with the revised version. Very happy that termination probability based on binomial distribution of after n th round of circular transcription corresponds to the linear one.